# A high-resolution, easy-to-build light-sheet microscope for subcellular imaging

John Haug[1,2], Seweryn Gałecki[1,2,3], Hsin-Yu Lin[1,2], Xiaoding Wang[1,2], Kevin M Dean[1,2]*

[1]Lyda Hill Department of Bioinformatics, UT Southwestern Medical Center, Dallas, United States; [2]Cecil H. and Ida Green Center for Systems Biology, UT Southwestern Medical Center, Dallas, United States; [3]Department of Systems Biology and Engineering, Silesian University of Technology, Gliwice, Poland

 Assessment

This **valuable** study presents Altair-LSFM, a well-documented implementation of a light-sheet fluorescence microscope (LSFM) designed for accessibility and reduced cost. The approach provides **compelling** evidence of its strengths, including the use of custom-machined baseplates, detailed assembly instructions, and demonstrated live-cell imaging capabilities. This manuscript will be of interest to microscopists and potentially biologists seeking accessible LSFM tools.

*For correspondence:
Kevin.Dean@UTsouthwestern.
edu

## Abstract

Although several open-source, easy-to-assemble light-sheet microscope platforms already exist—such as mesoSPIM, OpenSPIM, and OpenSpin—they are optimized for imaging large specimens and lack the resolution required to visualize subcellular features, such as organelles or cytoskeletal architectures. In contrast, lattice light-sheet microscopy (LLSM) achieves the resolution necessary to resolve such fine structures but, in its open-source implementation, can be alignment- and maintenance-intensive, often requiring specialist expertise. To address this gap, we developed Altair light-sheet fluorescence microscopy (LSFM), a high-resolution, open-source, sample-scanning light-sheet microscope specifically designed for subcellular imaging. By optimizing the optical pathway in silico, we created a custom baseplate that greatly simplifies alignment and assembly. The system integrates streamlined optoelectronics and optomechanics with seamless operation through our open-source software, *navigate*. Altair-LSFM achieves lateral and axial resolutions of approximately 235 and 350 nm, respectively, across a 266 μm field of view after deconvolution. We validate the system's capabilities by imaging sub-diffraction fluorescent nanospheres and visualizing fine structural details in mammalian cells, including microtubules, actin filaments, nuclei, and Golgi apparatus. We further demonstrate its live-cell imaging capabilities by visualizing microtubules and vimentin intermediate filaments in actively migrating cells.

## Introduction

Light-sheet fluorescence microscopy (LSFM) has revolutionized volumetric imaging by enabling rapid, minimally invasive 3D investigations of diverse biological specimens (*Huisken et al., 2004*). By illuminating the sample with a thin sheet of light from the side and capturing two-dimensional (2D) images in a highly parallel format, LSFM dramatically reduces photobleaching and out-of-focus blur. Although the optical foundations of LSFM were established as early as 1903 and later adapted for cleared specimens (*Voie et al., 1993*), it was the demonstration of LSFM on living, developing embryos in 2004 (*Huisken et al., 2004*) that triggered a wave of innovation, leading to specialized variants tailored to diverse biological contexts. Collectively, these methods have facilitated the long-term tracking of cells

through embryological development (*Lange et al., 2024*), the mapping of brain architecture (*Gao et al., 2019*) and activation patterns (*Ahrens et al., 2013*), and much more, making LSFM indispensable for a wide array of dynamic, 3D imaging applications.

While these milestones underscore LSFM's transformative potential, it wasn't until the early 2010s that researchers harnessed LSFM for subcellular imaging (*Planchon et al., 2011*). Achieving this level of detail requires optimization of both resolution and sensitivity, parameters fundamentally governed by the numerical aperture (NA) of the microscope objectives used. In LSFM, lateral resolution depends on the fluorophore's emission wavelength and the NA of the detection objective, whereas photon collection efficiency scales with the square of the detection NA. Consequently, high-NA objectives are essential for resolving fine, low-abundance biological structures while maximizing signal collection. Axial resolution, in turn, is set by the detection objective's depth of focus and the thickness of the illumination beam. When the illumination beam is thinner than the depth of focus, its thickness defines the axial resolution (*Gao et al., 2014*); conversely, when the beam is thicker than the depth of focus, the depth of focus defines the axial resolution, and fluorescence elicited outside this region contributes to the image as blur. Importantly, there is a trade-off between field of view, axial resolution, and NA: pushing for high axial resolution often constrains the accessible field of view, necessitating careful mechanical and optical design choices.

Bounded by these fundamental constraints, several methods have been developed to achieve subcellular resolution in LSFM. For example, one can illuminate the specimen with a propagation-invariant beam (*Planchon et al., 2011*) or an optical lattice (*Chen et al., 2014*). This can be done coherently, as in lattice light-sheet microscopy (LLSM) (*Chen et al., 2014*), or incoherently, as in field synthesis (*Chang et al., 2019*). However, the four-beam 'square' optical lattice, which was used in 16 of the 20 figure subpanels in the original LLSM study, was later found to provide little measurable improvement in resolution or sectioning compared to a traditional Gaussian beam (*Chang et al., 2020*). Another approach, dual view inverted selective plane illumination microscopy (diSPIM), captures images from multiple orthogonal perspectives and computationally fuses them using iterative deconvolution, significantly improving axial resolution (*Wu et al., 2013*). However, this method requires precise image registration and intensive computational processing. Axially swept light-sheet microscopy (ASLM) (*Dean et al., 2015*) extends the field of view while maintaining high axial resolution but operates at lower speeds and sensitivity compared to LLSM and diSPIM, making it less suitable for fast volumetric imaging. Oblique plane microscopy (OPM) offers another alternative by imaging an obliquely launched light sheet with a non-coaxial and complex optical train, allowing for single-objective light-sheet imaging but introducing substantial alignment challenges (*Sapoznik et al., 2020*). While these techniques offer powerful solutions for subcellular imaging, they all require expert assembly and routine alignment, limiting their widespread adoption. Turnkey commercial variants, such as the ZEISS Lattice Lightsheet 7, offer automated operation and high stability but remain costly and allow limited end-user modifiability. As a result, there remains a critical need for a high-resolution, accessible LSFM system that combines state-of-the-art imaging performance with straightforward assembly, reproducibility, and lower cost.

To address these limitations, we developed Altair-LSFM, a high-resolution, open-source light-sheet microscope that achieves subcellular detail while remaining accessible and easy to use. Altair-LSFM is built upon two guiding optical principles. First, in LLSM, the sole improvement in lateral resolution comes from the use of a higher-NA detection objective, which we incorporate to maximize both resolution and photon collection efficiency. Second, when diffraction effects are fully accounted for, a tightly focused Gaussian beam achieves a beam waist and propagation length that is comparable to that of a square lattice, eliminating the need for specialized optical components while preserving high axial resolution. By leveraging these principles, Altair-LSFM delivers optical performance on par with LLSM but without the added design complexity of LLSM. To streamline assembly and ensure reproducibility, we designed the optical layout for Altair-LSFM in silico, enabling a predefined optical alignment with minimal degrees of freedom. A custom-machined baseplate with precisely positioned dowel pins locks optical components into place, minimizing degrees of freedom and removing the need for fine manual adjustments. Additionally, by simplifying the optomechanical design and integrating compact optoelectronics, Altair-LSFM reduces system complexity, making advanced light-sheet imaging more practical for a wider range of laboratories.

**Table 1.** Light-sheet fluorescence microscopy (LSFM) variants and their associated illumination and detection optics.
The table lists the type of microscope, the illumination, and detection optics—including numerical aperture (NA) where available and immersion type in parentheses—as well as the overall design architecture (e.g. rail carrier, cage system, etc.).

| Microscope | Illumination optics | Detection optics | Design |
|---|---|---|---|
| OpenSPIM (*Pitrone et al., 2013*) | Olympus UMPLFLN 10× W NA 0.3 (Water) | UMPLFLN 20× W NA 0.5 (Water) | Rail Carrier |
| X-OpenSPIM (*Girstmair et al., 2022*) | Nikon CFI Plan Fluor 10× NA 0.3 (Water) | Nikon CFI Apochromat NIR 40× W NA 0.8 (Water) | Rail Carrier |
| EduSPIM (*Jahr et al., 2016*) | Zeiss LSFM 5× NA 0.1 (Air) | Zeiss LD Epiplan 5× NA 0.13 (Air) | Cage System |
| OpenSPIN (*Gualda et al., 2013*) | Nikon CFI Plan Fluor 4× NA 0.13 (Air) Nikon Plan Fluor 10× NA 0.3 (Air) | Nikon CFI Plan Fluor 4× NA 0.13 (Air) Nikon CFI75 LWD 16× NA 0.8 (Water) | Rail Carrier |
| UC2 (*Diederich et al., 2020*) | Generic 4× NA 0.14 (Air) | Generic 4× NA 0.14 (Air) Generic 10× NA 0.3 (Air) | 3D Printed Components |
| pLSM (*Chen et al., 2024*) | Mitutoyo Plan Apo 10× NA 0.28 (Air) | Mitutoyo Plan Apo 10× NA 0.28 (Air) ASI 54-10-12 16.67× NA 0.4 (Multi-Immersion) | Cage System |
| descSPIM (*Otomo et al., 2024*) | 500 and 150 mm cylindrical lenses (Air) | Thorlabs TL2X-SAP 2× NA 0.1 (Air) | Cage System |
| mesoSPIM (*Voigt et al., 2019*) | Nikon 50 mm f/1.4 G (Air) | Olympus MVPLAPO 1× NA 0.15 (Air) | Rail Carrier |
| BT-mesoSPIM (*Vladimirov et al., 2024*) | Nikon 50 mm f/1.4 G (Air) | Variable. Magnification 2–20×, NA 0.1–0.28 (Air) | Cage System |
| diSPIM (*Wu et al., 2013*) | Nikon CFI Apochromat NIR 40× W NA 0.8 (Water) | Nikon CFI Apochromat NIR 40× W NA 0.8 (Water) | Dovetail Tube System |
| CompassLSM (*Liu et al., 2021*) | Olympus XLFLUOR 4× NA 0.28 (Air/Water) | Olympus MVX PLAPO 1× NA 0.25 (Air) Olympus MVX PLAPO 2× C NA 0.5 (Air) Olympus UPlanFL 4× NA 0.13 (Air) Nikon CFI Plan Apo 10× C NA 0.5 (Water) | Rail Carrier |
| Lattice Light-Sheet Microscopy (LLSM) (*Chen et al., 2014*) | Special Optics 54-10-7 28.6× NA 0.67 (Water) | Nikon CFI75 Apochromat 25× C NA 1.1 (Water) | Free Space Optics |
| Altair-LSFM | Thorlabs TL20X-MPL 20× NA 0.6 (Water) | Nikon CFI75 Apochromat 25× C NA 1.1 (Water) | Custom Baseplate & Dovetail Tube System |

By combining high-resolution imaging with an accessible and reproducible design, Altair-LSFM addresses a critical gap in LSFM—bringing subcellular imaging capabilities to a broader scientific community. Its reliance on fundamental microscopy principles rather than overly complex optical systems ensures both performance and simplicity, while its modular architecture allows for straightforward assembly and operation. By eliminating the need for specialized optics and intricate alignment procedures, Altair-LSFM significantly lowers the barrier to adoption, making advanced light-sheet imaging feasible for laboratories that lack the resources or expertise to implement more complex systems. This combination of performance, accessibility, and scalability establishes Altair-LSFM as a powerful and practical solution for a wide range of laboratories.

## Results
### Survey of open-source LSFM designs
Before designing Altair-LSFM, we first evaluated existing open-source LSFM implementations to identify common design features, constraints, and trade-offs (*Table 1*). Many systems, such as UC2 (*Diederich et al., 2020*), pLSM (*Chen et al., 2024*), and EduSPIM (*Jahr et al., 2016*), were explicitly developed with cost-effectiveness in mind, relying on low-cost components and simplified designs to maximize accessibility. Others, including OpenSPIM (*Girstmair et al., 2022*; *Pitrone et al., 2013*),

OpenSPIN (*Gualda et al., 2013*), and mesoSPIM (*Vladimirov et al., 2024*; *Voigt et al., 2019*), were optimized for imaging large specimens, such as developing embryos or chemically cleared tissues. Most of these systems employed modular construction methods based on rail carriers or cage systems, which, while reducing alignment complexity compared to free-space optics, still retain degrees of freedom that can lead to misalignment and increase setup difficulty. Moreover, these microscopes generally operate at low magnification and low NA, limiting their ability to resolve subcellular structures. Among the surveyed designs, only diSPIM (*Wu et al., 2013*; *Kumar et al., 2014*) was explicitly developed for subcellular imaging, built with a dovetail-based system for precise optical alignment. However, diSPIM's most widely deployed configuration uses NA 0.8 objectives, which limits its photon collection efficiency and resolution. Consequently, the cell biology community lacks an open-source light-sheet microscope that combines state-of-the-art resolution with ease of assembly, robust optical alignment, and streamlined computational processing.

## Design principles of Altair-LSFM

Building on these findings, we designed Altair-LSFM to achieve performance comparable to the open-source variant of LLSM (*Chen et al., 2014*) while maintaining a compact footprint and streamlined assembly. Although cost-effectiveness was an important consideration throughout the design process, achieving sensitive, high-resolution imaging necessitated the use of high-NA optics, precision stages, stable laser sources, and high-performance, low-noise, high-quantum-efficiency cameras, all of which inherently increase system cost. To streamline procurement and integration, we minimized the number of required manufacturers while maintaining high-performance components. Including our optical table and laser source, the estimated price for Altair-LSFM is $150,000. A detailed list of all system components, their sources, and associated costs is provided in *Supplementary file 1* and *Supplementary file 2*, and a broader discussion of the design trade-offs, including the rationale for proprietary versus open-source hardware, and associated cost-benefit considerations, is provided in Appendix 1, Supplementary note 1.

This initial iteration of Altair-LSFM is specifically designed for imaging thin, adherent cells on 5 mm glass coverslips in aqueous media (n~1.33). For imaging such specimens, a sample-scanning approach is preferred over a light-sheet-scanning approach, as it minimizes the optical path length through the specimen, enabling use of more tightly focused illumination beams that improve axial resolution (*Figure 1—figure supplement 1*). If optical tiling is employed, Altair-ASLM could also be used for imaging expansion microscopy specimens (*Gao et al., 2019*). While Altair-LSFM could be used for superficial imaging in semitransparent embryos, systems implementing multiview illumination and detection schemes are generally better suited for such specimens (*McDole et al., 2018*). Similarly, cleared tissue imaging typically requires high-refractive index media (~1.45–1.56) and solvent-compatible objectives, along with methods such as ASLM or diSPIM that decouple the trade-off between field of view and axial resolution (*Chakraborty et al., 2019*; *Guo et al., 2020*).

Altair-LSFM is configured with a detection path that is nearly identical to that of LLSM, ensuring similar resolution (~230 nm×230 nm×370 nm) and photon collection efficiency. Specifically, it includes a 25× NA 1.1 water-dipping physiology objective (Nikon N25X-APO-MP) with a 400 mm achromatic tube lens (Applied Scientific Instrumentation), ensuring Nyquist sampling (130 nm pixel size) across the full width of a standard 25 mm CMOS camera (Hamamatsu Orca Flash 4.0 v3), yielding a total field of view of 266 µm. Emission filters were positioned in the focusing space immediately before the camera, and the entire detection assembly was built around a dovetail-based tube system to ensure robust alignment and mechanical stability (*Figure 1a*). To facilitate precise axial positioning and accommodate different sample types, the entire detection assembly was mounted on a 50 mm travel focusing stage.

With the detection path establishing the necessary criteria for resolution, field of view, and optical alignment (e.g. beam height), we next designed the illumination system. In LSFM, the foci of the illumination and detection objectives must precisely overlap without mechanical interference, limiting the choice of compatible objectives. To meet these requirements, we selected a 20× NA 0.6 long-working distance water immersion objective (Thorlabs TL20X-MPL). The spacing between this combination of illumination and detection objectives limits the size of usable coverslips to 5 mm, a constraint that is shared by the original LLSM design. While handling and mounting small coverslips can be challenging, we addressed this by designing a custom-machined coverslip holder to streamline the

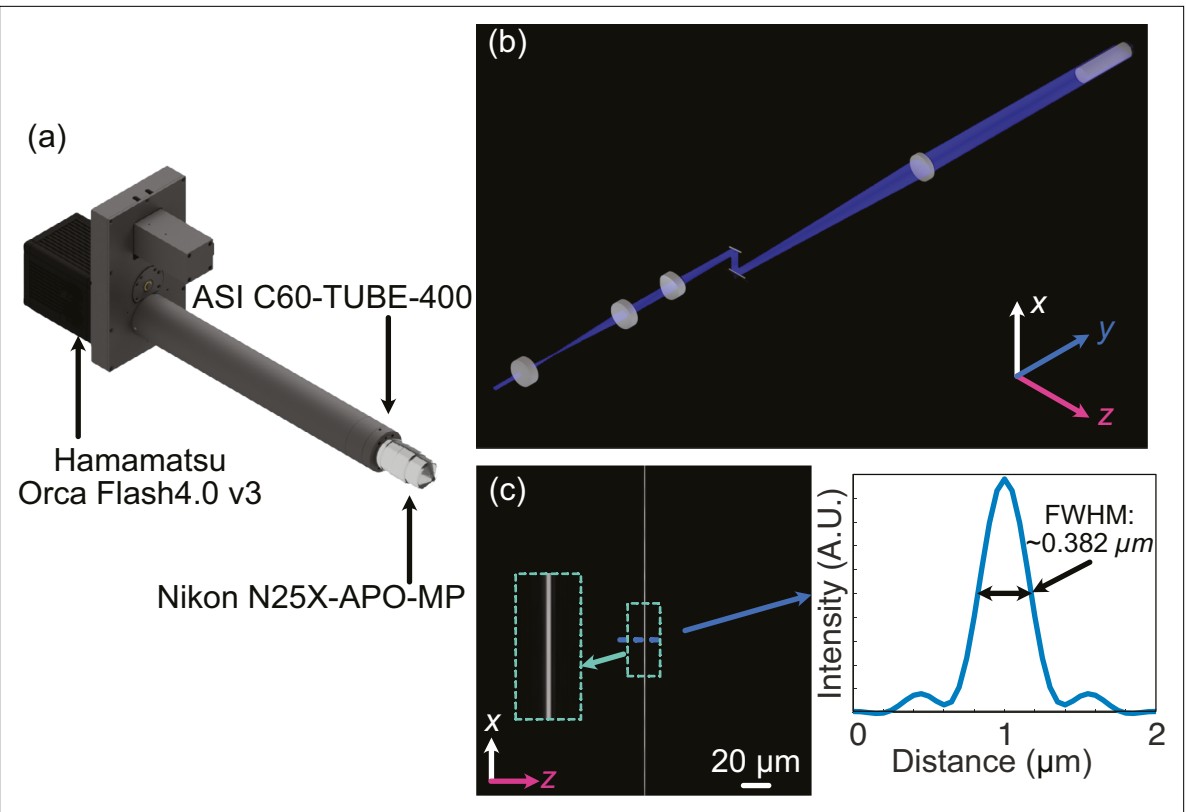

**Figure 1.** Optical design of Altair light-sheet fluorescence microscopy (LSFM). (**a**) Rendering of the detection arm elements. (**b**) Zemax Opticstudio layout and beam path of optimized illumination arm, where L1, L2, L3, and L4 are 30, 80, 75, and 250 mm achromatic doublets, respectively, and ILO is the TL20X-MPL illumination objective. (**c**) The simulated light-sheet beam profile in the xz plane at the focus of the illumination objective. The inset shows an enlarged region of the illumination light sheet, highlighting light-sheet thickness and uniformity. (**d**) The cross-sectional profile through the center of the light-sheet beam profile in (**c**), where the full-width-half-maximum (FWHM) of the light sheet was found to be 0.382 μm.

The online version of this article includes the following figure supplement(s) for figure 1:

**Figure supplement 1.** Comparison of sample- and light-sheet-scanning modes.

**Figure supplement 2.** Educational illustration depicting the conceptual function of the resonant galvo unit.

**Figure supplement 3.** Tolerance analysis of illumination optics.

mounting process. For users wishing to accommodate larger coverslips, the Nikon 25× objective can be substituted with a Zeiss W Plan-Apochromat 20×/1.0, whose slimmer form factor allows the co-focus between the illumination and detection objectives to occur beyond the physical body of the lenses, enabling the use of virtually any coverslip size (*Moore et al., 2021*). Guided by these constraints, we selected optical components capable of generating a theoretically diffraction-limited light sheet using straightforward magnification calculations.

The complete illumination system was designed for a collimated input beam with a 2 mm diameter, which first passes through an achromatic doublet lens (f=30 mm) and then a second achromatic doublet (f=80 mm) that expands and re-collimates it. After expansion, the beam passes through a rectangular aperture before reaching an achromatic cylindrical lens (f=75 mm). The rectangular aperture is conjugate to the back pupil plane of the cylindrical lens, enabling precise tuning of the light sheet's NA, and consequently its thickness and propagation length. This adjustability allows optimization of the light sheet for specimens of different thicknesses. The cylindrical lens focuses the beam in one direction to form the initial light sheet, and its focal length was chosen to provide sufficient spacing between optical elements for practical assembly. The shaped beam is then directed onto a resonant galvanometer, which improves illumination uniformity by rapidly pivoting the light sheet to average out shadowing artifacts arising from scattering and absorption within the sample (conceptual example shown in *Figure 1—figure supplement 2*; *Huisken and Stainier, 2007*; *Ricci et al., 2022*). After reflection from a 45° tilted mirror, the beam is relayed through an achromatic doublet

(f=250 mm) before entering the back aperture of the illumination objective, where it is finally focused onto the sample. This optical arrangement ensures a well-defined, dynamically pivoted light sheet that provides uniform illumination while mitigating shadowing effects.

## In silico optimization of Altair-LSFM

To ensure optimal illumination performance, we modeled the full illumination pathway of Altair-LSFM in Zemax OpticStudio (Ansys), systematically optimizing the relative placement of every optical element to achieve the desired focusing and collimation properties (*Figure 1b*). Each lens was iteratively adjusted to minimize aberrations and ensure precise beam shaping, enabling the formation of a well-defined light sheet. The design was centered around a 488 nm illumination wavelength, with spatial axes defined following standard conventions: the Y-dimension represents the laser propagation direction, Z corresponds to the detection axis, and X is orthogonal to both. The final illumination system, depicted in *Figure 1b*, was optimized to generate a diffraction-limited light sheet with a full-width-half-maximum (FWHM) of ~0.385 μm in Z, spanning the full 266 μm field of view, as shown in *Figure 1c and d*.

Beyond idealized modeling, designing a physically realizable system requires an understanding of how fabrication tolerances affect optical performance. To assess system robustness, we performed a tolerance analysis, which quantifies sensitivity to mechanical perturbations. This analysis evaluates how small positional or angular deviations of optical elements—caused by manufacturing imperfections—impact key performance metrics such as light-sheet thickness and displacement from the ideal position, allowing us to systematically evaluate system stability (*Figure 1—figure supplement 3a*). The perturbations analyzed were based on standard machining tolerances, typically ±0.005 in, with higher-precision machining achievable at ±0.002 in at increased cost. Given that Altair-LSFM was designed assuming the use of Polaris mounts, which incorporate DIN-7m6 ground dowel pins to aid with alignment, we considered the impact of angular misalignments caused by dowel pin positioning errors. In the worst-case scenario—where one dowel pin was offset by +0.005 in and the other by –0.005 in—the resulting angular deviation was ~1.45° (*Figure 1—figure supplement 3b*). To further assess system resilience, we conducted Monte Carlo simulations incorporating these perturbations, simulating a range of misalignment scenarios to quantify their effect on light-sheet performance. Our results showed that finer machining tolerances resulted in a worst-case performance closer to the nominal system, as visualized in *Figure 1—figure supplement 3c*, which compares the nominal, best, and worst configurations. Notably, the analysis identified that angular offsets in the galvo mirror had the most significant impact on light-sheet quality, highlighting the importance of tighter machining tolerances for this component to maximize system stability and performance.

## Optomechanical design of Altair-LSFM

Based on our simulation results, standard machining tolerances were deemed sufficient to construct a custom baseplate that ensures robust alignment and a compact, plug-and-play design. Unlike cage- or rail-based systems, custom baseplates minimize variability by enforcing a fixed spatial relationship between optical components, enabling assembly by nonexperts. Where possible, we eliminated all unnecessary degrees of freedom, restricting manual adjustments to only a few critical components. Specifically, we retained laser collimation (tip/tilt/axial position), galvo rotation, folding mirror alignment (tip/tilt), and objective positioning (tip/tilt/axial position).

To translate the numerically optimized positions of each optical element into a manufacturable design, the coordinates were imported into computer-aided design (CAD) software (Autodesk Inventor), ensuring precise positioning of all associated optomechanics. Where possible, Polaris optical posts and mounts were used to maintain consistency in mounting schemes and element heights. For components where a commercially available Polaris-compatible mount did not exist, such as the rotation mount (Thorlabs RSP1) for the cylindrical lens and the horizontal aperture (Thorlabs VA100), custom adapters were developed to seamlessly integrate them into the system. This approach allowed us to account for the offset between the optical element and its mechanical mount, ensuring that the baseplate was precisely machined with dowel pin locations and mounting holes (*Figure 2a*, *Figure 2—figure supplement 1*). Additionally, the baseplate features four mounting holes at its corners, spaced such that it can be directly secured to an optical table or elevated using additional posts, allowing for easy adjustment of the illumination path height. This modular, precision-engineered design is meant

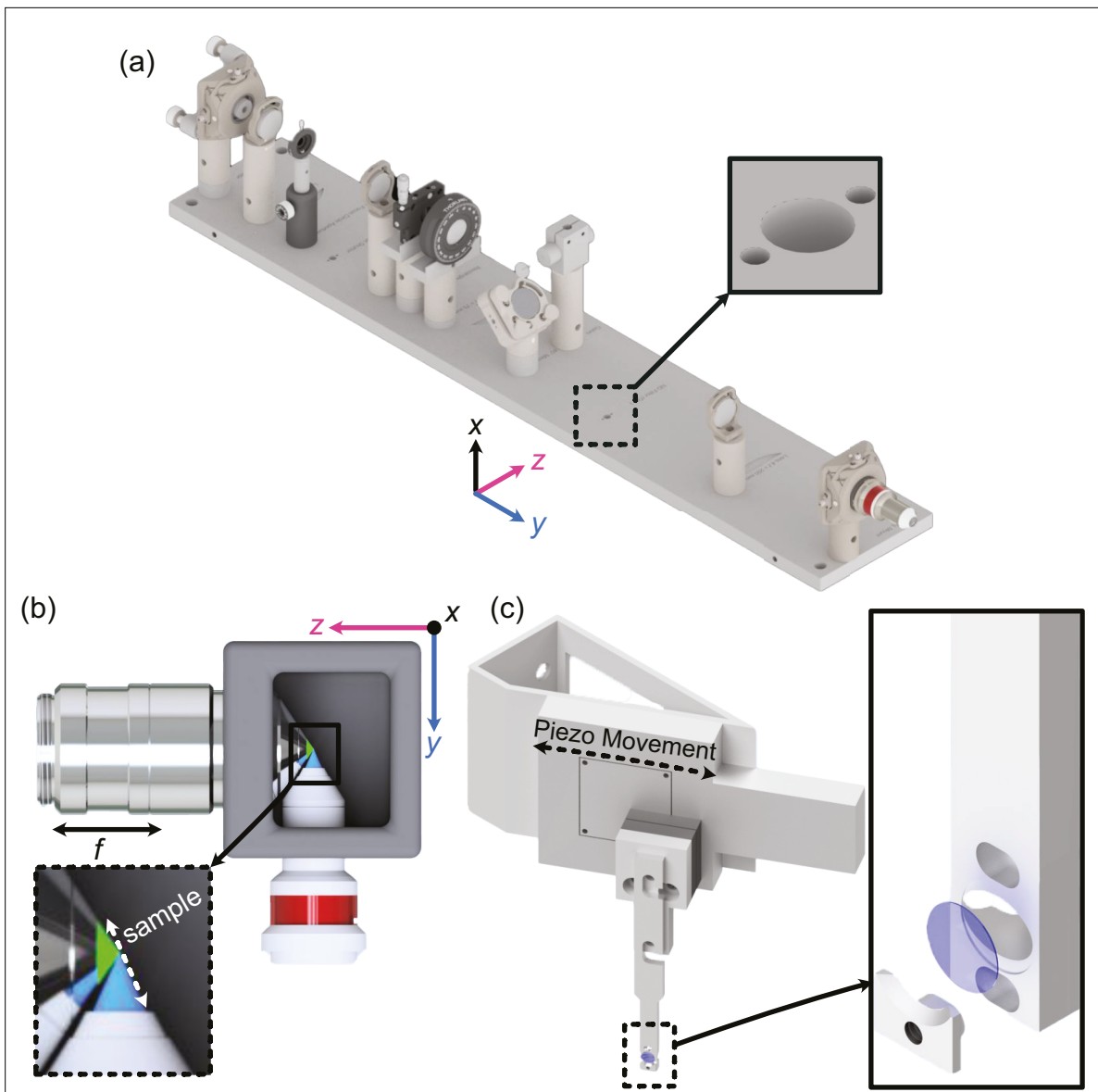

**Figure 2.** Mechanical design of Altair light-sheet fluorescence microscopy (LSFM). (**a**) Rendering of the completed illumination arm baseplate, with an inset showing the dowel pin holes compatible with the Polaris mounting line from Thorlabs. (**b**) Overhead view of the imaging configuration of our system, where our detection objective and illumination objectives are placed orthogonal to each other and the sample is scanned diagonally in the space between them in the axial direction shown by the white dashed line. (**c**) Rendering of our sample mounting and translation system. Here, a piezo motor is mounted onto an angled adapter to allow precise translation over the diagonal region between the objectives. Our custom 5 mm coverslip sample holder is also featured, where the inset shows an exploded assembly of the holder.

The online version of this article includes the following figure supplement(s) for figure 2:

**Figure supplement 1.** Baseplate post-installation procedure.

**Figure supplement 2.** Computer-aided design (CAD) model of the Altair light-sheet fluorescence microscopy (LSFM) system.

**Figure supplement 3.** Custom sample chamber in Altair light-sheet fluorescence microscopy (LSFM).

**Figure supplement 4.** System wiring diagram for optoelectronic control.

to ensure both ease of use and general mechanical stability, enabling integration of Altair-LSFM into alternative experimental setups.

Beyond the illumination path, the full microscope system is visualized in *Figure 2—figure supplement 2*. It incorporates a variety of translational and custom mounting elements to facilitate precise sample positioning and stable imaging. A sample chamber was designed and 3D printed to match

the working distances and clearances of the chosen illumination and detection objectives, offering two port configurations: one for traditional orthogonal imaging and another linear configuration that allows direct imaging of the light sheet itself (*Figure 2b*, *Figure 2—figure supplement 3*). Each port is equipped with two sets of O-rings, creating a liquid-proof seal around the objectives while still permitting smooth translation of the detection objective for focusing. Additionally, the snug fit of the O-rings naturally guides the user toward proper positioning of the illumination and detection objectives, decreasing the likelihood of alignment errors. The sample positioning assembly consists of three motorized translation stages (Applied Scientific Instrumentation), enabling precise positioning of the sample in x, y, and z. To enable rapid z-stack acquisition, we designed an angled bracket (θ=29.5°) for mounting a high-speed piezo (PiezoConcept HS1.100), which attaches directly to the sample positioning stages (*Figure 2c*). The sample is secured using a custom-designed sample holder for 5 mm glass coverslips, which features a clam-based mechanism—the coverslip is placed within a circular recess and secured in place by a screw-down clamp. Due to the angled sample scanning configuration, our collected image stacks must undergo a deskewing operation. All custom component designs and deskewing software are available for download at https://thedeanlab.github.io/altair.

## Optoelectronic design of Altair-LSFM

In addition to simplifying the optomechanical design, we also streamlined the electronics and control architecture of Altair-LSFM to minimize complexity and improve system integration. To achieve this, we consolidated all control electronics into a single controller (TG16-BASIC, Applied Scientific Instrumentation), which manages the operation of all linear translation stages (X, Y, Z, and F), as well as the power supply for the resonant galvo and sample scanning piezo. This approach significantly reduces the number of auxiliary controllers and power supplies, simplifying the physical setup. Currently, all timing operations are performed using a 32-channel analog output device (PCIe-6738, National Instruments), which is responsible for generating the global trigger, controlling the camera's external trigger, modulating the laser through analog and digital signals, setting the piezo control voltage, and providing the DC voltage for adjusting the resonant galvo amplitude. The resonant galvo used for shadow reduction operates at 4 kHz, ensuring that it is not rate-limiting for any acquisition mode described here. An overview of the electronics used in the system, along with an associated wiring diagram, is provided in *Figure 2—figure supplement 4* and *Supplementary file 3*.

All control electronics are operated through *navigate*, our open-source light-sheet microscope control software, which integrates hardware coordination, waveform generation, and data acquisition within a unified software environment (*Marin et al., 2024*). The combined performance of the control electronics and *navigate* defines the system's maximum temporal resolution. Mechanically, the acquisition of a z-stack is constrained by the response time of the sample-scanning piezo. Approximating the piezo as a first-order system gives a characteristic response time of ~0.35/f (seconds), where f is the actuator's resonant frequency (*Franklin and Emami-Naeini, 2019*). For our piezo (HS100, PiezoConcept), this gives an ideal response time of ~0.23 ms; however, accounting for additional physical considerations such as the weight of a sample holder, we expect this value to realistically be on the order of 1–5 ms for small step sizes. We also evaluated the rate at which a z-stack could be acquired with *navigate* using representative settings for coverslip-mounted cells (50μm z-stack, 0.25 μm step size, camera field of view of 512×2048). With a 10 ms exposure time, the system achieved image acquisition rates as fast as 62.5 Hz, with an average dead time of ~7.25 ms due to camera readout and piezo stepping (*Supplementary file 4*). Moreover, as demonstrated previously, the data-writing performance of *navigate* varies slightly depending on imaging parameters (e.g. number of z-slices and time points, owing to metadata overhead), with write speeds surpassing 1 gigavoxel/s under optimal conditions (*Marin et al., 2024*). Consequently, the integrated hardware and control software establish a unified, optoelectronic platform that balances performance, stability, and accessibility for advanced light-sheet applications.

## Alignment and characterization of Altair-LSFM

To evaluate optical performance, we first assembled and aligned the Altair-LSFM illumination system. The fiber-coupled laser source (Oxxius L4CC), which provides four excitation wavelengths (405, 488, 561, and 638 nm), was introduced and collimated using tip/tilt mounts. The collimated beam was then directed onto the resonant galvo, which was rotated to reflect the beam downward toward the

optical table. From there, the folding mirror was adjusted to guide the beam along the optical axis of the remaining components. Finally, the lateral position of the illumination objective was fine-tuned to ensure coaxial back-reflections, completing the alignment process. With the optical path aligned, we proceeded to validate system performance by characterizing the generated light sheet, where visualization of the light sheet is accomplished by a solution of fluorescein in transmission. As shown in *Figure 3a*, the light-sheet focus spans the full 266 µm field of view, closely matching our simulation results. Cross-sectional analysis of the FWHM in the z-dimension, presented in *Figure 3b*, reveals a z-FWHM of ~0.415 µm.

To assess the system's resolution, we imaged 100 nm fluorescent beads. Our image-processing pipeline involves first deskewing our acquired volumetric image stack using custom Python routines, and then deconvolution via PetaKit5D (*Ruan et al., 2024*) (see Materials and methods). The need for shearing arises when the scan axis does not align with the optical detection axis, as is the case for LLSM and Altair-LSFM when operating in a sample-scanning format, as well as both sample-scanning and laser-scanning OPMs. The point spread function of a single isolated fluorescent bead is shown in *Figure 3c–e*. The Gaussian-fitted distribution of FWHM measurements, performed on a population of fluorescent beads across a z-stack, is shown in *Figure 3f*. Prior to deconvolution, the average FWHM values measured across the bead population were 328 nm in x, 330 nm in y, and 464 nm in z. After deconvolution with PetaKit5D, these values improved to 235.5 nm in x, 233.5 nm in y, and 350.4 nm in z, achieving our desired resolution goals for subcellular imaging.

## Subcellular imaging with Altair-LSFM

To demonstrate the imaging capabilities of Altair-LSFM in a biological context, we prepared and imaged mouse embryonic fibroblast (MEF) cells stained for multiple subcellular structures. The staining protocol enabled visualization of the nucleus (DAPI, 405 nm, cyan), microtubules (488 nm, gray), actin filaments (561 nm, gold), and the Golgi apparatus (638 nm, magenta), corresponding to the excitation channels of our system. Deconvolved maximum-intensity projections of the labeled cells are shown in *Figure 4a and f*, with each individual corresponding to fluorescence channel presented in *Figure 4g–j*. The imaging results reveal fine nucleolar features within the nucleus, with perinuclear Golgi structures distinctly visible. Additionally, stress fibers are clearly resolved in the actin channel, and individual microtubules appear well defined, highlighting the system's ability to capture cytoskeletal structures with high resolution. These results confirm that Altair-LSFM provides the subcellular resolution, optical sectioning, and multicolor imaging performance necessary for quantitative biological imaging applications. To further assess performance in live specimens, we demonstrated dual-channel live-cell imaging. We modified the sample chamber to provide temperature control (see Appendix 1, Supplementary note 2, and *Figure 4—figure supplement 1*) and imaged retinal pigment epithelial (RPE) cells with endogenously GFP and TagRFP-T-tagged microtubules and vimentin, respectively (*Animation 1*, *Figure 5*). The cells exhibited robust motility, with time-lapse sequences revealing continuous, cell-side reorganization of microtubule and vimentin intermediate filaments throughout their bodies. Together, the fixed- and live-cell results establish that Altair-LSFM supports high-contrast, multicolor volumetric imaging at subcellular resolution in both static and dynamic cellular contexts.

## Discussion

In this work, we demonstrated the viability of a baseplate-based approach for the dissemination of a high-performance light-sheet microscope that is both accessible and straightforward to assemble by nonexperts. By combining optical simulations, precision-machined component design, and experimental validation, we developed Altair-LSFM, a system that delivers subcellular resolution imaging with minimal alignment requirements. Characterization using fluorescent beads confirmed that Altair-LSFM achieves a resolution of 328, 330, and 464 nm before deconvolution, which improves to ~235 and 350 nm after deconvolution, in XY, and Z, respectively. These values are on par with the original, open-source version of LLSM (~230 and 370 nm, in XY and Z, respectively; *Chen et al., 2014*), confirming that our approach achieves state-of-the-art performance but in a streamlined, cost-effective, and optically less complex format. For example, the open-source LLSM illumination path includes approximately 29 optical components, each requiring precise lateral, angular, and coaxial alignment and maintenance. In contrast, Altair-LSFM contains only nine such elements. By this metric, Altair-LSFM

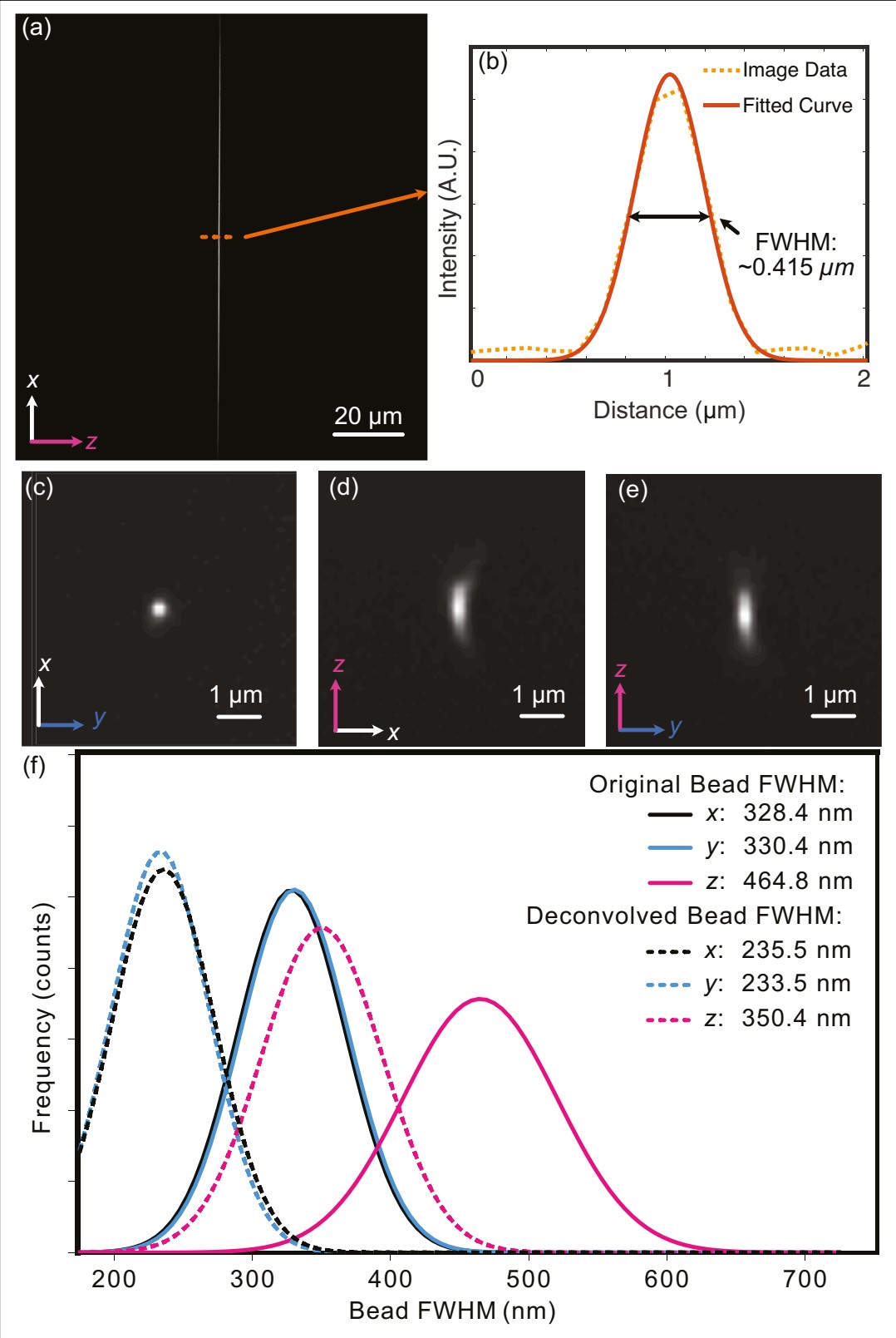

**Figure 3.** Experimental characterization of light-sheet thickness and resolution. (**a**) Experimental light-sheet beam profile at the focus. The light sheet was visualized in a transmission geometry with fluorescence derived from a fluorescein solution. (**b**) The center cross-section profile of (**a**), showing both raw data and a fitted curve with a full-width-half-maximum (FWHM) of ~0.415 µm. (**c–e**) Maximum-intensity projections for an isolated 100 nm fluorescent bead. All three orthogonal perspectives are provided to reveal any potential optical aberrations. A slight degree of coma is observable in the XZ view. (**f**) Gaussian-fitted distribution of the FWHM of beads imaged in a z-stack in each dimension both before (solid) and after (dashed) deconvolution.

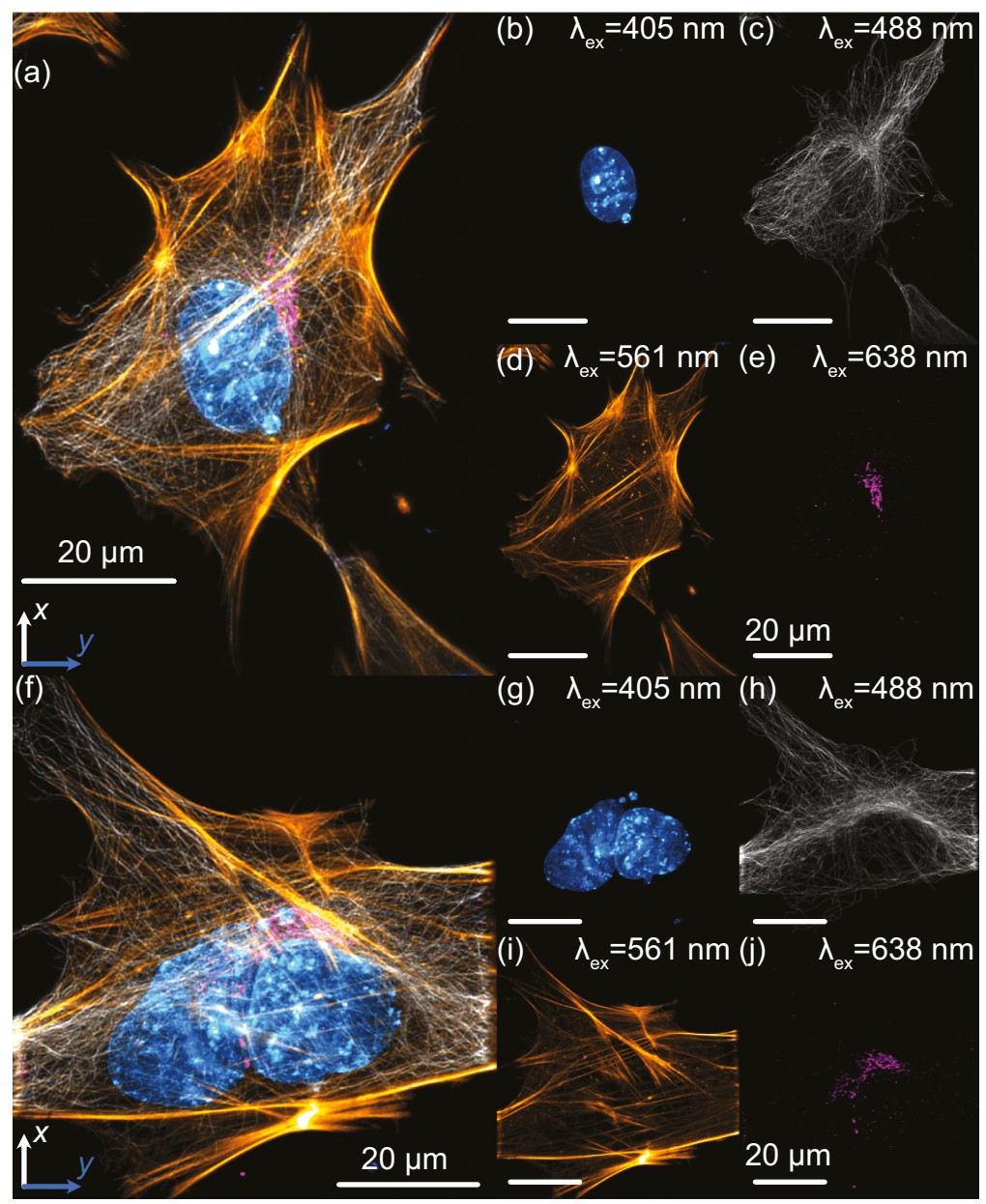

**Figure 4.** Multicolor subcellular imaging with Altair light-sheet fluorescence microscopy (LSFM). Lateral maximum-intensity projections of mouse embryonic fibroblasts (MEFs) fluorescently labeled with nuclei (cyan), tubulin (gray), actin (gold), and the Golgi apparatus (magenta). (**a**) Maximum-intensity projection of the full z-stack in the xy plane. (**b–e**) Individual channels corresponding to (**a**): (**b**) nuclei, (**c**) microtubules, (**d**) actin, and (**e**) Golgi apparatus. (**f**) Maximum-intensity projection of a second z-stack in the xy plane. (**g–j**) Individual channels corresponding to (**f**): (**g**) nuclei, (**h**) microtubules, (**i**) actin, and (**j**) Golgi apparatus. Nuclei were labeled with DAPI, actin filaments with phalloidin, and both microtubules and the Golgi apparatus were stained using indirect immunofluorescence.

The online version of this article includes the following figure supplement(s) for figure 4:

**Figure supplement 1.** Computer-aided design (CAD) renderings of custom heated sample chamber design.

is considerably simpler to assemble and maintain, further supporting our overarching goal of making high-resolution LSFM systems accessible to nonspecialist laboratories. Nonetheless, it is worth noting that the original LLSM system offered a greater number of illumination modes (e.g. square, hexagonal, and structured illumination) and supported imaging in both a sample- and light-sheet scanning configurations (Appendix 1, Supplementary note 3). A complete parts list and estimated cost are provided in

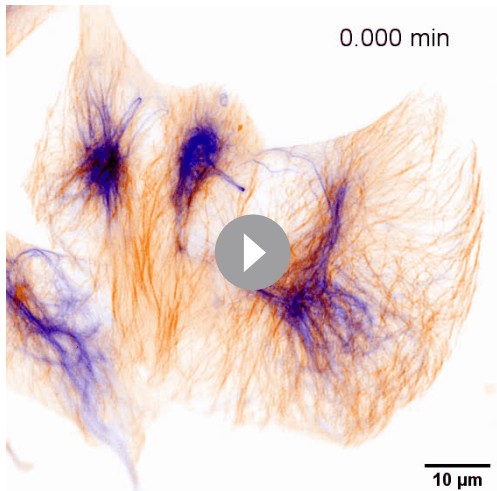

0.000 min

10 µm

**Animation 1.** Time-lapse of retinal pigment epithelial immortalized with human telomerase reverse transcriptase (RPE-hTERT) cells with endogenously tagged microtubules and vimentin. Time-lapse sequence of RPE-hTERT cells with endogenously tagged cytoskeletal markers: microtubules (red) and vimentin (blue).

*Supplementary file 1* and *Supplementary file 2*, respectively, offering a transparent roadmap for users looking to adopt and build the system. To aid planning, *Supplementary file 5* summarizes expected build and validation times, stratified by prior experience with optical system assembly and operation.

A number of open-source designs have also extended LSFM into the subcellular regime, including implementations of LLSM and diSPIM. These platforms have been instrumental in advancing dissemination and training within the imaging community, though they typically require routine optical alignment and operation by dedicated personnel. Commercial LLSM instruments from 3i and ZEISS have increased the availability of LLSM and have contributed substantially to disseminating this technology. According to vendor materials, the ZEISS variant of LLSM provides automated operation and long-term stability, providing a user-friendly, turnkey solution for subcellular imaging. The system's design, which incorporates a meniscus lens to enable oblique imaging through a coverslip, simplifies setup and usability, albeit with a modest reduction in achievable resolution (reported deconvolved resolution: ~290 nm×450–900 nm) relative to the

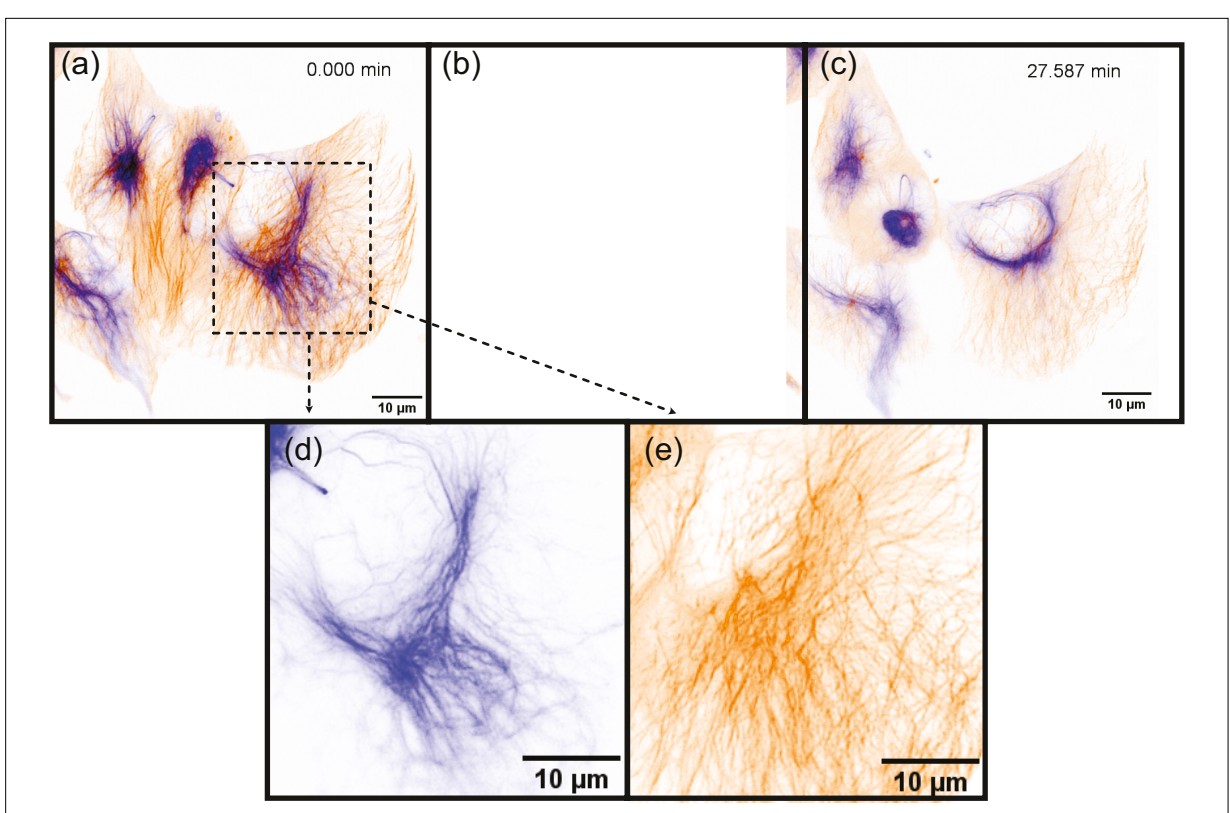

**Figure 5.** Live-cell volumetric imaging of intermediate filament and microtubule dynamics. Selected time points from a time-lapse sequence of actively migrating retinal pigment epithelial (RPE) cells, showing vimentin (blue) and microtubules (orange). The series comprises 50 frames; representative time points are displayed.

original LLSM implementation and Altair-LSFM. However, acquisition and service costs remain high, and system modification by end users is limited. Altair-LSFM addresses a different need: an openly documented, modifiable, and reproducible path to subcellular LSFM that nonspecialist laboratories can build, customize, and maintain at a fraction of the cost of a commercial system.

Building on this successful prototype, future efforts will focus on expanding both imaging capabilities and user experience. One natural evolution of this approach is the development of more advanced light-sheet microscope designs, such as ASLM (*Dean et al., 2015*; *Dean et al., 2022*) and OPM (*Sapoznik et al., 2020*; *Chang et al., 2021*), which offer additional flexibility for imaging a broader range of sample types. Notably, OPM avoids the challenges associated with nonstandard sample mounting and, owing to its single objective architecture, is fully compatible with standard environmental chambers used for live-cell imaging. Another avenue for improvement is the optimization of Altair-LSFM for cleared-tissue imaging, further extending its applicability into tissue contexts. Additionally, we aim to eliminate the need for an external analog output device, consolidating all triggering and waveform generation within a single unified controller, which will further reduce hardware dependencies and enhance system efficiency.

Another important consideration is the long-term scalability and routine maintenance of Altair-LSFM in a variety of lab environments. Multi-site benchmarking and community feedback will be pivotal to ensure consistent, reproducible performance across different setups. To aid planning, we also provide guidance on data storage in Appendix 1, Supplementary note 4. Future enhancements—such as self-alignment routines—could further boost imaging quality and throughput. To accelerate widespread adoption, we have thoroughly documented the entire assembly process in our GitHub repository (*Dean et al., 2025*), which is also provided as *Supplementary file 6*. Although the construction of custom optical systems may seem intimidating to nonexperts, our dissemination strategy draws inspiration from other successful open-source projects such as mesoSPIM, which has seen widespread adoption, including >30 implementations worldwide, through a similar model of exhaustive documentation, open-source control software, and community support via user meetings and workshops. For expert users who wish to tailor the instrument, we also provide all Zemax illumination-path simulations and CAD files, along with step-by-step optimization protocols, enabling modification and re-optimization of the optical system as needed. These customization resources are intended for users with prior experience in optical and optomechanical design, while the default configuration remains turnkey for nonexperts.

A persistent challenge in advanced microscopy is the lengthy interval—from instrument conceptualization to commercialization—often spanning close to a decade. By fostering open collaboration and continuous software-hardware integration, we envision Altair-LSFM evolving into a robust, ever-improving platform, well suited to meet future research challenges in high-resolution, volumetric imaging. Integrating these systems with *navigate* (*Marin et al., 2024*), we aim to further democratize intelligent imaging workflows and broaden the reach of cutting-edge instrumentation (Appendix 1, Supplementary note 5). With high-NA imaging, reduced mechanical and optical complexity, and lower cost, Altair-LSFM stands poised to accelerate LSFM adoption, delivering a powerful yet accessible solution for researchers immediately seeking access to high-resolution, cutting-edge, volumetric imaging.

## Materials and methods
### Acquisition and simulation computer
All microscope control and optical simulations were performed on a Colfax SX6300 workstation, configured to handle high-speed data acquisition and processing. It is equipped with dual Intel Xeon Silver 4215R processors (8 cores, 16 threads, 3.2 GHz), 256 GB of DDR4 3200 MHz ECC RAM, a 7.68 TB Samsung PM9A3 NVMe SSD for high-speed data acquisition, a 20 TB Seagate Exos X20 HDD for long-term data storage, a PNY NVIDIA T1000 4 GB GPU, and an Intel X710-T2L dual-port 10GbE.

### Cell culture
All cells were obtained from ATCC and routinely tested for mycoplasma contamination. Cells were maintained at 37°C in a humidified incubator with 5% $CO_2$, cultured 5 mm coverslips (pre-rinsed with 70% ethanol) placed in multiwell plates. MEFs were cultured in Dulbecco's modified Eagle's medium

(DMEM, Gibco) supplemented with 10% fetal bovine serum (FBS, Gibco) and 100 µg/mL penicillin-streptomycin. RPE immortalized with human telomerase reverse transcriptase (RPE-hTERT) Vimentin-GFP/mTubulin-RFP cells were generated by TALEN-based genome editing (*Gan et al., 2016*) and grown in ATCC-formulated DMEM/F12 supplemented with 10% FBS and 1% antibiotic-antimycotic. For imaging, cells were seeded on 5 mm round #0 coverslips placed in six-well plates. During imaging, the coverslip was secured in a chamber and bathed in prewarmed media. RPE-hTERT cells were imaged in DMEM/F12 medium without phenol red, supplemented with 5% FBS and 1% Anti-Anti. All imaging procedures were performed at 37°C.

## Immunofluorescence

MEFs were cultured to approximately 50% confluency before processing. Cells were first rinsed with pre-heated (37°C) 1× phosphate-buffered saline (PBS), and then briefly permeabilized and fixed with preheated PEM buffer (80 mM PIPES, 5 mM EGTA, 2 mM $MgCl_2$, [pH: 6.8]), supplemented with 0.3% Triton-X and 0.125% glutaraldehyde for 30 s. A secondary fixation was then performed in preheated PEM buffer containing 2% paraformaldehyde for 15 min at 37°C. Following fixation, cells were washed three times with 1× PBS (2 min each). Unless otherwise indicated, all subsequent incubations were performed at room temperature with constant agitation. Residual aldehydes were quenched using 5 mM glycine for 10 min, after which the cells were blocked for 1 hr in 3% bovine serum albumin (BSA) and 0.01% Triton-X in 1× PBS. For indirect immunofluorescence, cells were incubated overnight at 4°C with primary antibodies diluted in staining buffer [1% BSA+0.01% Triton-X in 1× PBS]: mouse anti-α-Tubulin (Sigma-Aldrich, #T9026, 1:250, RRID:AB_11204167) and rabbit anti-GOLGA/GM130 (Proteintech, #11308-1-AP, 1:500, RRID:AB_2115327). After three washes with PBST (1× PBS containing 0.01% Triton-X, 2 min each), cells were incubated for 1 hr with secondary antibodies diluted in staining buffer: donkey anti-mouse CF488A (Biotium, #20014-1, 1:500, RRID:AB_10853131) and donkey anti-rabbit Alexa Fluor 647 (Thermo Fisher Scientific, #A-31573, 1:500, RRID:AB_2536183). Actin filaments were stained with phalloidin-CF568 (Biotium, #00044, 1:50) in 1× PBS for 1 hr. Finally, cells were incubated in DAPI nuclear dye (Thermo Fisher Scientific, #62248, 300 nM) in 1× PBS for 10 min. Samples were stored at 4°C in 1× PBS with 0.02% sodium azide ($NaN_3$) until imaging.

## Preparation of fluorescent bead samples

A 5 mm glass coverslip was placed inside a Petri dish, and approximately 100 µL of (3-Aminopropyl) triethoxysilane (APTS) at a concentration of 5 mM was applied to its surface. The APTS was incubated for 10–30 min to promote bead adhesion, after which the coverslip was lightly washed three times with deionized water. Fluorescent beads (Fluoresbrite YG Microspheres 0.10 µm, Polyscience, Inc) were diluted to the desired concentration (typically $10^{-3}$ or $10^{-4}$ for a normal distribution, $10^{-6}$ for a sparse distribution) and applied to the treated coverslip, where they were incubated for 2–20 min, with longer incubation times increasing bead density. Finally, the coverslip was lightly washed with deionized water to remove unbound beads before imaging.

## Image deskewing

After image acquisition, deskewing was performed to correct for the non-orthogonal scanning geometry between the piezo stage and the optical axes of the microscope. A custom Python script available on our GitHub repository was used to perform deskewing. The user provides the path to the image stack along with the relevant imaging parameters: z-step size, xy pixel size, and the deskew angle. In our configuration, the deskew angle corresponds to 90°–60.5°, where 60.5° represents the angle between the normal of the sample mount and the microscope's y-axis.

## Image deconvolution

Deconvolution was performed using PetaKit5D (*Ruan et al., 2024*) with standard operating parameters. All datasets were processed with a background level set to 100, and two iterations of the Optical Transfer Function Masked Wiener (OMW) deconvolution algorithm were applied. The Wiener parameter (alpha) was set to 0.005, the Optical Transfer Function Cumulative Threshold to 0.6, and the Hann window bounds to 0.8–1. No edge erosion was applied, and the resulting data were saved as 16-bit images. PSFs were simulated using the PSFGenerator (*Kirshner et al., 2013*) package according to the illumination wavelength and NA of the light sheet, and the emission wavelength and NA of the

detection objective. PSFs were modeled using the Richards & Wolf 3D Optical Model with a refractive index of 1.3333.

## Custom machining and fabrication

All metal components were machined from 6061-T6x aluminum by Protolabs or Xometry, adhering to standard machining tolerances of ±0.005 in. The original sample chamber was 3D printed using a Formlabs Form 3B resin printer with standard black resin. Our live-cell sample chamber was machined from 6061-T6x aluminum from Xometry. CAD files for all custom parts and procedures on how to place an order from Xometry using them are provided as a supplementary file, with up-to-date versions available on GitHub.

## Acknowledgements

The authors would like to thank Calvin Jones and Dr. Sophia Theodossiou (Boise State University) for their assistance in designing and printing the custom sample chamber, and Melissa Glidewell for her initial evaluation of optical tolerances. This work was supported by the National Institutes of Health (U54CA268072 and RM1GM145399).

## Additional information

### Competing interests

Kevin M Dean: K.M.D. is a co-inventor on a patent covering ASLM and a founder of Discovery Imaging Systems, LLC. K.M.D. also has consultancy agreements with 3i, Inc (Denver, CO, USA). The other authors declare that no competing interests exist.

### Funding

| Funder | Grant reference number | Author |
|---|---|---|
| National Institute of General Medical Sciences | RM1GM145399 | Kevin M Dean |
| National Cancer Institute | U54CA268072 | Kevin M Dean |

The funders had no role in study design, data collection and interpretation, or the decision to submit the work for publication.

### Author contributions

John Haug, Formal analysis, Validation, Investigation, Visualization, Methodology, Writing – original draft, Writing – review and editing; Seweryn Gałecki, Hsin-Yu Lin, Resources, Visualization, Writing – original draft, Writing – review and editing; Xiaoding Wang, Investigation, Methodology; Kevin M Dean, Conceptualization, Resources, Software, Supervision, Funding acquisition, Investigation, Visualization, Methodology, Writing – original draft, Project administration, Writing – review and editing

### Author ORCIDs

John Haug ⓘ https://orcid.org/0000-0001-8247-3271
Seweryn Gałecki ⓘ https://orcid.org/0000-0002-7020-0804
Kevin M Dean ⓘ https://orcid.org/0000-0003-0839-2320

Reviewer #1 (Public review): https://doi.org/10.7554/eLife.106910.3.sa1
Reviewer #2 (Public review): https://doi.org/10.7554/eLife.106910.3.sa2
Reviewer #3 (Public review): https://doi.org/10.7554/eLife.106910.3.sa3
Author response https://doi.org/10.7554/eLife.106910.3.sa4

# Additional files

## Supplementary files

MDAR checklist

Supplementary file 1. Detailed equipment list for Altair light-sheet fluorescence microscopy (LSFM).

Supplementary file 2. Approximate cost to build an Altair light-sheet fluorescence microscopy (LSFM).

Supplementary file 3. Electrical pinouts used on National Instruments PCIe-6738 data acquisition card. All analog and digital connections were made using a National Instruments SCB-68A shielded terminal block.

Supplementary file 4. Acquisition performance specifications.

Supplementary file 5. Approximate time to assemble Altair light-sheet fluorescence microscopy (LSFM).

Supplementary file 6. Detailed documentation for the assembly and operation of Altair light-sheet fluorescence microscopy (LSFM).

## Data availability

All documentation describing the design and assembly of Altair LSFM has been archived on Zenodo (https://doi.org/10.5281/zenodo.18060763). The imaging data and figure source content supporting the findings of this study are available on Zenodo (https://doi.org/10.5281/zenodo.18049379). Microscope control was performed using Navigate v0.1.0, archived on Zenodo (https://doi.org/10.5281/zenodo.18134804).

The following previously published datasets were used:

| Author(s) | Year | Dataset title | Dataset URL | Database and Identifier |
|---|---|---|---|---|
| Deam KM | 2025 | The DeanLab/altair: eLife Publication | https://doi.org/10.5281/zenodo.18060763 | Zenodo, 10.5281/zenodo.18060763 |
| Haug J, Haug J, Lin H-Y, Wang X, Dean K, Gałecki | 2025 | A High-Resolution, Easy-to-Build Light-Sheet Microscope for Sub-Cellular Imaging. | https://doi.org/10.5281/zenodo.18049379 | Zenodo, 10.5281/zenodo.18049379 |
| Dean KM, Wang A, Marin Z, Collison D, Jinlong L, Augustine J, Chen B, Stephan D, Veerapaneni S, Sheppard S, Easha S, Evolene P, Connor H, Ngo T, Nguyen TD, Shepherd D, conorhughmcfadden, mehr0096, vmcspadden, andrewjUTSW, 3vwylie, ATcHoneybee, arjutsw, Rapuris, juhelh, nng-thienphu, Johnhaug223, kayl102, Elepicos, Buckelew D | 2026 | TheDeanLab/navigate: Publication of Altair-LSFM | https://doi.org/10.5281/zenodo.18134804 | Zenodo, 10.5281/zenodo.18134804 |

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

# Appendix 1

## Supplementary note 1. Design rationale and cost considerations for Altair

A core design objective of Altair-LSFM was to develop a high-performance, open-source light-sheet microscope that is accessible to a broad community of users, including those without extensive optical design or engineering experience. While we prioritized cost-effectiveness wherever possible, several design choices were made to balance performance, reliability, and ease of adoption, rather than minimizing cost alone. Moreover, we sought to minimize the complexity of sourcing, configuring, and integrating components from disparate vendors by favoring consolidated, multipurpose hardware. Despite this, end users are free to customize the hardware configuration to suit their experimental needs. Different detection objectives, stages, filter wheels, or cameras can be readily incorporated, as *navigate* natively supports a broad range of hardware devices. New hardware can also be easily integrated through *navigate's* modular device interface, enabling users to expand functionality or adapt the system to specific equipment without extensive reconfiguration.

The estimated total cost of a complete Altair-LSFM system in its default configuration, including the optical table and laser source, is approximately $150,000. Although this may still represent a barrier for some research groups, it is substantially lower than most commercial systems offering comparable subcellular resolution performance (e.g. LLSM systems from 3i or Zeiss, which are ~$600,000 and >$800,000 USD, respectively, or a diSPIM from ASI, which is ~$250,000). These values vary substantially depending on system configuration and are provided only as approximate guides based on publicly available information, including instrumentation, published news articles, and personal communications. One major cost driver in LLSM-based systems is the use of a spatial light modulator, which is a relatively low-efficiency device and necessitates upgrading to higher-power laser sources (*Liu et al., 2023*) (e.g. ~500 mW). For Altair-LSFM, a full list of system components, vendors, and approximate costs is provided in *Supplementary file 1* and *Supplementary file 2*. Below, we outline the primary cost drivers in the system, our rationale for selecting them, and potential alternatives that could be used in future variants to reduce cost, albeit by potentially compromising system performance.

### Illumination and detection objectives

The illumination and detection objectives used in LSFM must be treated as a single, coupled optical system. For Altair-LSFM, the primary cost driver is the detection/illumination pair built around the Nikon N25X-APO-MP 25×/1.1 NA detection objective (≈ $33,000), which must be paired with either the Thorlabs TL20X-MPL 20×/0.6 NA illumination objective (≈$5000) or the Special Optics 54-10-7@488–910 nm (≈$15,000). The Special Optics objective provides a slightly higher NA (0.66 NA) but a shorter working distance (3.74 mm) compared to the Thorlabs TL20X-MPL (0.6 NA, 5.5 mm). Both are optimized for water immersion and achromatically corrected across the visible spectrum. Using the vectorial Richards & Wolf 3D optical model, the expected illumination PSF at 488 nm and maximum NA is approximately 451 nm×2240 nm (XY×Z) for the Special Optics 54-10-7 and 496 nm×2711 nm for the Thorlabs TL20X-MPL. While the higher NA of the Special Optics lens marginally improves resolution, the increased working distance of the Thorlabs objective makes it compatible with alternative detection objectives and facilitates imaging with larger coverslips (see below). Moreover, in practical applications, the full NA of these objectives is rarely used, as doing so would require exceptionally thin specimens.

Users seeking to reduce costs may substitute the Nikon 25× with a Zeiss W Plan-Apochromat 20×/1.0 (≈$7000). This objective is compatible with the existing detection path (no different tube lens required; aberration corrections are internal to the primary objective) and, when paired with the TL20X-MPL for illumination, is expected to deliver similar practical performance while easing sample handling by allowing coverslips larger than 5 mm in diameter. A more aggressive cost reduction is to redesign the optical train around Nikon N40X-NIR 40×/0.8 objectives for both illumination and detection. However, because vendor ray files/Zemax models are not available for these lenses, such a redesign would require introducing additional alignment degrees of freedom and tolerancing steps, increasing assembly complexity. Moreover, the lower NA would reduce raw resolution to ~400 nm laterally and axially (*Dean et al., 2015*) and halve photon collection, which scales with $NA^2$,

lowering sensitivity—trade-offs that may be unacceptable for low-signal or fast volumetric imaging. While the combination of objectives used in this implementation of Altair-LSFM has a total list price of approximately $38,000, substituting the detection objective with the Zeiss W Plan-Apochromat 20×/1.0 would reduce this cost to ≈ $12,000, and a full redesign around Nikon 40×/0.8 objectives for both illumination and detection would further reduce the total to ≈$6000, providing a clear path toward more cost-effective configurations depending on experimental needs and available resources.

## Scientific camera

Scientific CMOS (sCMOS) cameras are a major investment in Altair-LSFM. Older Hamamatsu ORCA-Flash 4.0 units could sometimes be obtained near ~$15k, whereas current-generation sCMOS from major vendors (Hamamatsu, Photometrics, PCO, etc.) typically fall in the $30k–$40k range. The cost is justified by characteristics that are especially important for light-sheet imaging: high quantum efficiency, low read noise with effective suppression of dark current, large sensors that enable highly parallel acquisition across a wide field, and fast readout rates that support high frame rates. In our configuration (50× effective magnification, 6.5 µm pixels), the camera provides Nyquist-sampled pixels (~130 nm) across a wide 25 mm sensor, which is critical for capturing ~150–200 planes per adherent cell per channel at biologically relevant speeds—particularly for live-cell experiments.

Lower-cost industrial CMOS cameras (e.g. Ximea MU196CR-ON, recently demonstrated in a Direct-View OPM [*Lamb et al., 2025*] configuration) can in principle substitute for sCMOS in budget-constrained builds. However, in our experience, these sensors exhibit higher noise floors and reduced dynamic range, which limits sensitivity in low-signal regimes. They are also typically slower than leading sCMOS cameras and show greater variability in fixed-pattern defects (e.g. hot pixels), often necessitating calibration and postprocessing. For applications that demand subcellular resolution at high speed and low illumination, we therefore recommend sCMOS as the default choice; industrial CMOS can be considered for cost-reduced variants with the understanding that overall performance will be compromised (sensitivity, speed, and image uniformity).

## ASI equipment

Altair-LSFM consolidates operation of all motion axes and the emission filter wheel into a single ASI Tiger Controller, including motorized XYZ sample positioning, a high-speed piezo Z scan for volumetric acquisition, and a motorized focus (F) axis for precise co-focusing of the detection plane. The complete system—XYZ and F stages, piezo stage, motorized emission filter wheel, ASI Tiger controller, and basic optomechanical adapters—totals ~$31,000 (~20% of system cost).

Cost-reduced variants are possible, but each entails performance or complexity trade-offs. Replacing motorized XYZ/F with manual stages can save ~$12,000, but removes critical capabilities such as autofocus, 3D tiling, and multi-position acquisition. Eliminating the piezo and relying solely on linearly encoded Z stages lowers upfront cost and complexity, yet significantly increases repositioning time and reduces volumetric throughput, which is especially limiting for live-cell imaging. Worth noting, our sample scanning piezo could deliver higher bandwidth with the vendor's dedicated controller; however, we favor a unified Tiger Controller-based control stack to minimize the number of controllers to wire, configure, and maintain, thereby streamlining adoption for nonexperts.

Open-source mechanical platforms (e.g. OpenFlexure; *McDermott et al., 2022*) could, in principle, be adapted to further reduce cost, but would require custom assembly and software integration—shifting burden to the end user—and may exhibit reduced accuracy/precision and increased hysteresis compared with the closed-loop stages used here. Similarly, the motorized emission filter wheel can be omitted in favor of a fixed multiband emission filter (saving ~$5000), but this increases spectral crosstalk and often necessitates post-acquisition spectral unmixing, complicating workflows and potentially reducing sensitivity for dim samples.

## Analog and digital control

Altair-LSFM uses a National Instruments (NI) 6738 (PXIe/PCIe form factor) for real-time analog and digital control. The board provides 32 analog outputs with 16-bit DAC resolution at up to 1 MS/s update rate, which we use to generate waveforms for the resonant/galvo drive, piezo position control, and analog laser intensity. The card also exposes 10 digital I/O lines that we use for laser modulation

(TTL) and camera triggering/synchronization. We selected this platform because it is readily available internationally, offers robust, deterministic timing, and comes with long-term driver support and tooling that reduces integration burden across operating systems and labs. In future releases, we aim to eliminate the NI card by consolidating triggering and waveform generation into the ASI Tiger Controller, further reducing hardware count, wiring, and software dependencies while keeping the system turnkey for nonexperts. Importantly, we opted to avoid lower-cost microcontroller (MCU) solutions (e.g. USB MCUs with external DACs) because they carry several practical disadvantages:

- Timing determinism and jitter: General-purpose MCUs and USB links typically exhibit µs-ms-scale jitter and nondeterministic latency, complicating tight synchronization between camera exposure, galvo phase, laser blanks, and piezo motion.
- Waveform fidelity and throughput: Many MCUs cannot sustain simultaneous, multichannel DAC streaming at 1 MS/s.
- Noise and analog performance: MCU-centric solutions often have higher analog noise, poorer calibration/drift, and require custom electrical interfaces.
- Scalability and maintenance: NI provides stable drivers, diagnostics, and a clearer path for multi-OS support.
- Feature trade-offs: Replacing the NI card would push more complexity into software/firmware and wiring.

## Supplementary note 2. Environment chamber design

To support live-cell imaging with Altair-LSFM, we developed a temperature-controlled sample chamber (*Figure 4—figure supplement 1*). Typical cell viability temperatures span ~25°C for yeast to 37°C for mammalian cells. The chamber integrates adhesive heating pads and embedded thermocouples in the chamber wall, regulated by a TempCo controller to maintain a fixed set point. Relative to our initial design, we removed the secondary illumination port (formerly used for linear light-sheet imaging), which allowed us to mount larger heaters along two exterior walls for more uniform heating of the chamber volume. To minimize thermal gradients—which can induce drift or optical aberrations (e.g. with the Nikon 25× objective), we also provide indirect objective heating. Two add-on hoods surround the illumination and detection ports; each wrapped with heating pads. In total, the system heats three zones: (1) illumination-objective hood, (2) detection-objective hood, and (3) the two external walls opposite the objective ports. All off-the-shelf components (outside the custom chamber) were sourced from McMaster-Carr; the total live-cell upgrade cost was ~$3250 (*Supplementary file 2*).

Commercial environmental chambers often maintain 5% $CO_2$ (stabilizing pH in bicarbonate-buffered media) and add humidification to limit evaporation; turnkey systems of this type typically cost on the order of $20,000. Such fully enclosed atmospheres are uncommon for light-sheet microscopes because multiple objectives contact the specimen, precluding a sealed enclosure, with the notable exception of a purpose-built enclosure reported previously (*McDole et al., 2018*). In the open-source implementation of LLSM, as well as the early LLSM variant commercialized by 3i, environmental control was achieved by enclosing both the specimen and the objectives in temperature-regulated blocks through which heated or chilled water was circulated, while pH stabilization was maintained using 10 mM HEPES buffer. However, humidity and $CO_2$ were not controlled in this configuration. We adopt a similar strategy in Altair-LSFM, using temperature-controlled components and HEPES-buffered media to maintain a stable imaging environment. Given our ~50 mL chamber volume, evaporation is minimal even during extended imaging sessions. In contrast, OPMs, and the more recent LLSM variant commercialized by ZEISS, are inherently compatible with commercially available environmental chambers and therefore avoid many of the practical constraints associated with multi-objective light-sheet systems.

## Supplementary note 3. Light-sheet illumination and acquisition modes

### LLSM illumination modes

The original implementation of LLSM, introduced by *Chen et al., 2014*, is a highly flexible platform capable of generating multiple illumination patterns, each optimized for different biological specimens. This flexibility arises from its use of a spatial light modulator to define the illumination pattern at the back focal plane of the illumination objective. The system can operate in a dithered,

or time-averaged square and hexagonal lattice modes, as well as in a structured illumination mode, providing a wide range of trade-offs between resolution, field of view, and imaging speed. However, the square lattice rapidly became the most widely used due to its ease of operation and compatibility with deconvolution workflows. Indeed, in the original 2014 publication, 16 of the 20 figure subpanels utilized the square lattice configuration, while only one panel was dedicated to each of the hexagonal or structured illumination modes.

The square lattice provides strong axial confinement with minimal side-lobe energy, producing uniform, high-contrast optical sectioning that is suitable for most cellular imaging applications. The hexagonal lattice, by contrast, improves axial resolution but is accompanied by side lobes that increase in intensity with laser propagation distance, limiting its use to particularly thin specimens. These side lobes also necessitate computational postprocessing for their removal, which becomes increasingly challenging when their intensity approaches ~50% of the main lobe, a condition that can occur even for modest beam propagation lengths (*Chang et al., 2020*). These beams can also operate in a structured illumination mode, wherein the lattice pattern is stepped or phase-shifted laterally to enhance optical resolution in both the X and Z dimensions. Because the orientation of the illumination pattern is fixed, resolution along the Y axis (e.g. the laser propagation direction) remains diffraction limited. Importantly, this mode requires five phase-shifted images per optical section to reconstruct a single high-resolution plane, substantially increasing both illumination dose and acquisition time. The computational reconstruction is also considerably more intensive than for standard lattice modes. Although structured illumination is supported in the commercial implementation of LLSM offered by 3i, its use in practice remains limited due to these experimental and computational demands.

The recently developed ZEISS Lattice Lightsheet 7 represents a streamlined and user-friendly evolution of the original LLSM design, prioritizing ease of operation over full optical flexibility. Unlike the earlier implementations, which support multiple lattice geometries and structured illumination modes, the ZEISS system offers three predefined 'Sinc' beam configurations. Each configuration corresponds to a specific trade-off between light-sheet length and thickness (e.g. 15×550 nm, 30×700 nm, and 100×1400 nm, each available with or without side lobes).

## Volume acquisition modes

To acquire volumetric data, both the original and 3i systems can operate in sample-scanning and light-sheet-scanning modes, whereas both Altair-LSFM and the ZEISS implementation of LLSM function exclusively in a sample-scanning format. In sample scanning, the specimen is translated through a stationary light sheet, whereas in light-sheet scanning, the light sheet is scanned in the Z-direction through a stationary specimen synchronously with the position of the detection objective (see *Figure 1—figure supplement 1*). Each mode offers distinct advantages and trade-offs. Sample scanning provides the simplest architecture to assemble and operate but requires shearing of the data to place the data in its proper spatial context. It is particularly advantageous for specimens with a large aspect ratio, such as highly adherent cells. In the light-sheet-scanning configuration, the sheet is scanned using a linear galvanometer conjugate to the back pupil of the illumination objective. This design necessarily introduces an additional optical fold into the illumination train, and the plane conjugate to the galvo must typically be relayed to the back pupil of the illumination objective—often with an f-theta lens and tube lens pair. The galvo sweep must also be calibrated to ensure precise synchronization between the light-sheet position and the detection objective's focal plane. Consequently, compared to sample scanning, light-sheet scanning increases both alignment complexity and cost. Unlike sample scanning, the acquired data are immediately in their correct spatial context, eliminating the need for postprocessing shearing. Light-sheet scanning is particularly useful for specimens with a more spherical aspect ratio. In the original LLSM publication, this mode was used to image biological processes such as microtubule and histone dynamics during mitosis, *Tetrahymena thermophila* swimming behavior, T-cell immunological-synapse formation, and *Caenorhabditis elegans* embryogenesis. It is also advantageous for vibration-sensitive specimens, such as neutrophil-like cells migrating within a compliant extracellular matrix, where sample translation could induce unwanted mechanical perturbations in the extracellular matrix (*Dean et al., 2016*).

## Resolution

While the original LLSM platform supports square, hexagonal, and structured illumination modes, our discussion on resolution here remains focused on the square lattice. Importantly, for both square lattice light sheets and Gaussian light sheets, axial resolution is tightly coupled to the propagation length of the illumination beam: Both square lattice light sheets, which do not behave as propagation-invariant beams, and Gaussian beams broaden in diameter as a function of propagation length. Consequently, like-for-like comparisons must match detection NA, sheet thickness at focus, light-sheet propagation length, and postprocessing.

The original LLSM work described lattice light sheets by inner/outer NA rather than by sheet thickness and propagation length, which complicates direct comparisons to Gaussian sheets. More recent reports with bead-based measurements (FWHM of 100 nm fluorophore beads) yielded a raw lateral and axial resolution of 312 nm × 666 nm for an excitation wavelength of 488 nm. For comparison, the original publication reported a resolution of ~230 nm × 370 nm, which is consistent with the use of shorter propagation length illumination beam and deconvolution. For comparison, the ZEISS implementation reports raw resolutions of 330 nm×500–1000 nm and deconvolved resolutions of 290 nm×450–900 nm, depending on the selected light-sheet configuration. The reduced resolution likely reflects the compromises inherent to imaging in an open-top geometry, which requires the use of a meniscus lens to accommodate the optical path. For Gaussian beams, a frequent source of confusion in the literature is the use of paraxial Gaussian beam formulae to estimate thickness and propagation length at NAs where diffraction effects are non-negligible. At the illumination NAs used in Altair-LSFM, scalar/vector diffraction modeling better predicts the realized beam properties than simple Gaussian optics. Thus, when propagation length and focus thickness are matched, a diffraction-limited Gaussian sheet can deliver performance comparable to a square lattice, without introducing side lobes.

Quantitative comparisons of resolution depend not only on the optics but also on the exact algorithmic implementation of deskewing (shearing) and deconvolution. For deconvolution, outcomes are sensitive to PSF quality, background estimation, regularization, and iteration count; moreover, most algorithms assume a nonaberrated, shift-invariant PSF, an assumption that degrades with specimen heterogeneity. In our measurements, Altair-LSFM achieves ~235 nm × 350 nm after deconvolution, comparable to the deconvolved values reported for square lattice LLSM under matched detection NA and sampling. These reported measurements were performed on 100 nm fluorescent beads and likely reflect illumination beams optimized for the thinnest biological specimens. Thicker specimens would require longer propagation lengths and correspondingly thicker light sheets, yielding resolutions closer to those previously reported—approximately 312 nm×666 nm before deconvolution or 230 nm×460 nm after deconvolution, assuming an ~30% reduction in the point spread function dimensions.

## Comparison with other light-sheet modalities

diSPIM attains near-isotropic resolution by imaging the same specimen from two orthogonal detection paths and fusing the volumes with multiview deconvolution. Reported performance ranges from ~330 nm isotropic in the original implementation (*Kumar et al., 2014*) to ~380 nm isotropic in commercial systems. For cleared-tissue variants of diSPIM using mixed-immersion 17.9×/0.4 NA objectives, single-view resolutions of ~840 nm (lateral) and 4600 nm (axial) have been improved to ~800 nm isotropic after registration and one iteration of deconvolution (*Guo et al., 2020*). In practice, diSPIM excels when large fields of view and improved axial sectioning are required and when multiview registration and deconvolution are acceptable parts of the workflow. High-resolution ASLM systems use aberration-free remote focusing to translate the light sheet along its propagation direction in synchrony with a camera's rolling shutter, effectively decoupling axial sectioning from field of view. Early implementations with ~0.8 NA detection achieved ~400 nm isotropic resolution (*Dean et al., 2015*); adaptations for high-RI cleared samples have reported ~330 nm isotropic resolution (*Chakraborty et al., 2019*). Large field of view variants such as the mesoSPIM (*Voigt et al., 2019*) use electro-tunable lenses to scan the illumination beam and report typical resolutions of ~2.5 µm (lateral) and ~5 µm (axial). Both diSPIM and ASLM excel when one needs to maximize the field of view while maintaining a high axial resolution.

Recent advances in OPM (*Yang et al., 2019*) have enabled subcellular imaging while preserving a single-objective, inverted geometry that is compatible with standard sample preparation, environmental chambers, and autofocus systems. A high-NA OPM has reported 299 nm×731 nm (raw) and ~209 nm×523 nm (after deconvolution), albeit with short working distance (*Sapoznik et al., 2020*). Longer-working-distance, water-dipping OPMs extend penetration and support optical tiling (e.g. volumes on the order of 800 µm×500 µm×200 µm) with raw resolutions around ~400 nm×1220 nm (*Chen et al., 2022b*). OPM also allows novel illumination approaches, such as DaXi, which used an image flipper to perform multiview imaging with ~450 nm lateral and ~2 µm axial resolution throughout an ~3000 µm×800 µm×300 µm volume. Likewise, by incorporating an image rotator into the optical path, structured illumination could be leveraged to improve the lateral resolution to ~140 nm (*Chen et al., 2022a*). And lastly, because of the shared optical train, a single deformable mirror is capable of simultaneously correcting aberrations in both the illumination and detection (*Mcfadden et al., 2024*). Given the rapid evolution of OPMs in particular, and LSFM more broadly, the comparisons provided here are intended to situate Altair-LSFM in context, not to offer an exhaustive review.

## Supplementary note 4. Data storage and handling

Storage costs are approximate and are expected to vary widely between institutions, depending on negotiated contracts, available infrastructure, and funding models. The cost estimates provided here are intended solely as a practical guide for planning purposes and should not be considered definitive or universally applicable. For example, large cloud providers list prices that vary from roughly $0.023 to $0.00099 per GB per month, with additional monitoring, retrieval, and transfer fees that vary by storage class. In contrast, the local computing infrastructure at UT Southwestern Medical Center charges a one-time fee of $300 per TB, which is amortized over 5 years. Advantageously, to reduce costs to end users, our local file system includes an automated 100 PB tape archive: data inactive for more than 12 months are transparently migrated while remaining visible as placeholders and are automatically recalled when accessed. Our lab operates with a 130 TB quota, which costs ~$8000 per year. In general, we typically budget $2k per year per experimentalist for storage and compute, rising to $10–12k per year for users that rely on high-end GPUs. These expenses are routinely incorporated into federally funded grant budgets. To operate within the confines of such a quota, we recommend several simple rules to manage data overhead.

- Save data in a Zarr format, which is supported by *navigate* and uses lossless blosclz compression by default.
- Immediately assess data for quality. *Navigate* automatically saves maximum-intensity projections that can be reviewed quickly in Fiji/ImageJ on standard computers, enabling rapid assessment of data quality. Evaluating data in 2D is substantially faster than loading full 3D stacks; datasets of insufficient quality should be deleted immediately to prevent unnecessary storage costs.
- Store only the raw image data. Saving postprocessing routines, such as deconvolution, increases the storage footprint. Instead, we reprocess data on demand using high-performance tools such as PetaKit5D (*Ruan et al., 2024*). Importantly, to support reproducibility, one must record processing parameters such as deconvolution iterations and point spread functions alongside each dataset to enable faithful reprocessing.

## Supplementary note 5. Comparison of *navigate* with other open-source microscope control software

A wide variety of open-source software packages are available for microscope control, the most popular of which is Micro-Manager (*Edelstein et al., 2010*), a highly extensible and widely adopted platform. Both *navigate* and Micro-Manager provide user-friendly graphical interfaces and support image-based feedback control, making them accessible to a broad community of users. Micro-Manager's greatest strength lies in its extensive library of device drivers, which supports a vast array of commercial hardware from multiple manufacturers. Although Micro-Manager has been successfully used to control light-sheet microscopes, it typically requires manual hardware configuration and custom scripting to implement advanced or nonstandard acquisition workflows.

In contrast, *navigate* was developed from the ground up to provide turnkey support for multiple light-sheet architectures, including both sample-scanning and light-sheet-scanning LSFM configurations, as well as ASLM and OPM. While its hardware library is more focused than Micro-Manager's, *navigate* was developed in tandem with the Altair-LSFM hardware, enabling tightly integrated, preconfigured acquisition routines optimized for this system. This co-design minimizes setup time, eliminates the need for user-side customization, and ensures reliable synchronization across devices. The result is a software environment that provides robust performance, efficient data handling, and streamlined operation for nonexpert users.

*Navigate* also offers advanced data management capabilities through native support for multiple file formats, including TIFF, HDF5, N5, and Zarr. Each file is saved with embedded Open Microscopy Environment (OME) metadata and is fully compatible with BigDataViewer, facilitating interoperability with downstream image analysis pipelines. Additionally, *navigate's* modular Python-based architecture allows expert users to extend functionality, integrate new devices, and leverage the extensive Python ecosystem for intelligent or adaptive acquisition workflows.

Although *navigate* is provided as the default control platform for Altair-LSFM, the system can, in principle, be configured to operate with Micro-Manager or other open-source control frameworks. We welcome and encourage community-led efforts to build such compatibility. Ultimately, our goal with *navigate* is to provide a ready-to-use, flexible, and extensible control environment that lowers the barrier to advanced light-sheet imaging while supporting continued community innovation and customization.

