## [Editor Report · eLife Assessment]

This **valuable** study presents Altair-LSFM, a well-documented implementation of a light-sheet fluorescence microscope (LSFM) designed for accessibility and reduced cost. The approach provides **compelling** evidence of its strengths, including the use of custom-machined baseplates, detailed assembly instructions, and demonstrated live-cell imaging capabilities. This manuscript will be of interest to microscopists and potentially biologists seeking accessible LSFM tools.

---

## [Referee Report · Reviewer #1 (Public review)]

Summary:

The article presents the details of the high-resolution light-sheet microscopy system developed by the group. In addition to presenting the technical details of the system, its resolution has been characterized and its functionality demonstrated by visualizing subcellular structures in a biological sample.

Strengths:

The article includes extensive supplementary material that complements the information in the main article.

Live imaging has been incorporated, as requested, increasing the value of the paper.

Weaknesses:

None

---

## [Referee Report · Reviewer #2 (Public review)]

Summary:

The authors present Altair-LSFM (Light Sheet Fluorescence Microscope), a high-resolution, open-source light-sheet microscope, that may be relatively easy to align and construct due to a custom-designed mounting plate. The authors developed this microscope to fill a perceived need that current open-source systems are primarily designed for large specimens and lack sub-cellular resolution or achieve high-resolution but are difficult to construct and are unstable. While commercial alternatives exist that offer sub-cellular resolution, they are expensive. The authors manuscript centers around comparisons to the highly successful lattice light-sheet microscope, including the choice of detection and excitation objectives. The authors thus claim that there remains a critical need for a high-resolution, economical and easy to implement LSFM systems and address this need with Altair.

Strengths:

The authors succeed in their goals of implementing a relatively low cost (~ USD 150K) open-source microscope that is easy to align. The ease of alignment rests on using custom-designed baseplates with dowel pins for precise positioning of optics based on computer analysis of opto-mechanical tolerances as well as the optical path design. They simplify the excitation optics over Lattice light-sheet microscopes by using a Gaussian beam for illumination while maintaining lateral and axial resolutions of 235 and 350 nm across a 260-um field of view after deconvolution. In doing so they rest on foundational principles of optical microscopy that what matters for lateral resolution is the numerical aperture of the detection objective and proper sampling of the image field on to the detection, and the axial resolution depends on the thickness of the light-sheet when it is thinner than the depth of field of the detection objective. This concept has unfortunately not been completely clear to users of high-resolution light-sheet microscopes and is thus a valuable demonstration. The microscope is controlled by an open-source software, Navigate, developed by the authors, and it is thus foreseeable that different versions of this system could be implemented depending on experimental needs while maintaining easy alignment and low cost. They demonstrate system performance successfully by characterizing their sheet, point-spread function, and visualization of sub-cellular structures in mammalian cells including microtubules, actin filaments, nuclei, and the Golgi apparatus.

Weaknesses:

There is still a fixation on comparison to the first-generation lattice light-sheet microscope, which has evolved significantly since then:

(1) One of the major limitations of the first generation LLSM was the use of a 5 mm coverslip, which was a hinderance for many users. However, the Zeiss system elegantly solves this problem and so does Oblique Plane Microscopy (OPM), while the Altair-LSFM retains this feature which may dissuade widespread adoption. This limitation and how it may be overcome in future iterations is now discussed in the manuscript but remains a limitation in the currently implemented design.

(2) Further, on the point of sample flexibility, all generations of the LLSM, and by the nature of its design the OPM, can accommodate live-cell imaging with temperature, gas, and humidity control. In the revised manuscript the authors now implement temperature control, but ideal live cell imaging conditions that would include gas and humidity control are not implemented. While, as the authors note, other microscopes that lack full environmental control have achieved widespread adoption, in my view this still limits the use cases of this microscope. There is no discussion on how this limitation of environmental control may be overcome in future iterations.

(3) While the microscope is well designed and completely open source it will require experience with optics, electronics, and microscopy to implement and align properly. Experience with custom machining or soliciting a machine shop is also necessary. Thus, in my opinion it is unlikely to be implemented by a lab that has zero prior experience with custom optics or can hire someone who does. Altair-LSFM may not be as easily adaptable or implementable as the authors describe or perceive in any lab that is interested even if they can afford it. Claims on how easy it may be to align the system for a "Novice" in supplementary table 5, appear to be unsubstantiated and should be removed unless a Novice was indeed able to assemble and validate the system in 2 weeks. It seems that these numbers were just arbitrarily proposed in the current version without any testing. In our experience it's hard to predict how long an alignment will take for a novice.

(4) There is no quantification on field uniformity and the tunability of the light sheet parameters (FOV, thickness, PSF, uniformity). There is no quantification on how much improvement is offered by the resonant and how its operation may alter the light-sheet power, uniformity and the measured PSF.

---

## [Referee Report · Reviewer #3 (Public review)]

Summary:

This manuscript introduces a high-resolution, open-source light-sheet fluorescence microscope optimized for sub-cellular imaging.

The system is designed for ease of assembly and use, incorporating a custom-machined baseplate and in silico optimized optical paths to ensure robust alignment and performance.

The important feature of the microscope is the clever and elegant adaptation of simple gaussian beams, smart beam shaping, galvo pivoting and high NA objectives to ensure a uniform thin light-sheet of around 400 nm in thickness, over a 266 micron wide Field of view, pushing the axial resolution of the system beyond the regular diffraction limited-based tradeoffs of light-sheet fluorescence microscopy.

Compelling validation using fluorescent beads multicolor cellular imaging and dual-color live-cell imaging highlights the system's performance. Moreover, a very extensive and comprehensive manual of operation is provided in the form of supplementary materials. This provides a DIY blueprint for researchers that want to implement such a system, providing also estimate costs and a detailed description of needed expertises.

Strengths:

- Strong and accessible technical innovation.

With an elegant combination of beam shaping and optical modelling, the authors provide a high resolution light-sheet system that overcomes the classical light-sheet tradeoff limit of thin light-sheet and small field of view. In addition, the integration of in silico modelling with a custom-machined baseplate is very practical and allows for ease of alignment procedures. Combining these features with the solid and super-extensive guide provided in the supplementary information, this provides a protocol for replicating the microscope in any other lab.

- Impeccable optical performances and ease of mounting of samples

The system takes advantage of the same sample-holding method seen already in other implementations, but reduces the optical complexity. At the same time, the authors claim to achieve similar lateral and axial resolution to Lattice-light-sheet microscopy although without a direct comparison (see below in the "weaknesses" section). The optical characterization of the system is comprehensive and well-detailed. Additionally, the authors validate the system imaging sub-cellular structures in mammalian cells.

-Transparency and comprehensiveness of documentation and resources.

A very detailed protocol provides detailed documentation about the setup, the optical modeling and the total cost.

Conclusion:

Altair-LSFM represents a well-engineered and accessible light-sheet system that addresses a longstanding need for high-resolution, reproducible, and affordable sub-cellular light-sheet imaging. At this stage, I believe the manuscript makes a compelling case for Altair-LSFM as a valuable contribution to the open microscopy scientific community.

Comments on revisions:

I appreciate the details and the care expressed by the authors in answering all my concerns, both the bigger ones (lack of live cell imaging demonstration) and to the smaller ones (about data storage, costs, expertise needed, and so on). The manuscript has been greatly improved, and I have no other comments to make.

---

## [Author Response]

The following is the authors’ response to the original reviews.

**eLife Assessment**
This useful study presents Altair-LSFM, a solid and well-documented implementation of a light-sheet fluorescence microscope (LSFM) designed for accessibility and cost reduction. While the approach offers strengths such as the use of custom-machined baseplates and detailed assembly instructions, its overall impact is limited by the lack of live-cell imaging capabilities and the absence of a clear, quantitative comparison to existing LSFM platforms. As such, although technically competent, the broader utility and uptake of this system by the community may be limited.

We thank the editors and reviewers for their thoughtful evaluation of our work and for recognizing the technical strengths of the Altair-LSFM platform, including the custom-machined baseplates and detailed documentation provided to promote accessibility and reproducibility. Below, we provide point-by-point responses to each referee comment. In the process, we have significantly revised the manuscript to include live-cell imaging data and a quantitative evaluation of imaging speed. We now more explicitly describe the different variants of lattice light-sheet microscopy—highlighting differences in their illumination flexibility and image acquisition modes—and clarify how Altair-LSFM compares to each. We further discuss challenges associated with the 5 mm coverslip and propose practical strategies to overcome them. Additionally, we outline cost-reduction opportunities, explain the rationale behind key equipment selections, and provide guidance for implementing environmental control. Altogether, we believe these additions have strengthened the manuscript and clarified both the capabilities and limitations of AltairLSFM.

**Public Reviews:**

**Reviewer #1 (Public review):**
Summary:The article presents the details of the high-resolution light-sheet microscopy system developed by the group. In addition to presenting the technical details of the system, its resolution has been characterized and its functionality demonstrated by visualizing subcellular structures in a biological sample.Strengths:(1) The article includes extensive supplementary material that complements the information in the main article.(2) However, in some sections, the information provided is somewhat superficial.

We thank the reviewer for their thoughtful assessment and for recognizing the strengths of our manuscript, including the extensive supplementary material. Our goal was to make the supplemental content as comprehensive and useful as possible. In addition to the materials provided with the manuscript, our intention is for the online documentation (available at thedeanlab.github.io/altair) to serve as a living resource that evolves in response to user feedback. We would therefore greatly appreciate the reviewer’s guidance on which sections were perceived as superficial so that we can expand them to better support readers and builders of the system.

Weaknesses:(1) Although a comparison is made with other light-sheet microscopy systems, the presented system does not represent a significant advance over existing systems. It uses high numerical aperture objectives and Gaussian beams, achieving resolution close to theoretical after deconvolution. The main advantage of the presented system is its ease of construction, thanks to the design of a perforated base plate.

We appreciate the reviewer’s assessment and the opportunity to clarify our intent. Our primary goal was not to introduce new optical functionality beyond that of existing high-performance light-sheet systems, but rather to substantially reduce the barrier to entry for non-specialist laboratories. Many open-source implementations, such as OpenSPIM, OpenSPIN, and Benchtop mesoSPIM, similarly focused on accessibility and reproducibility rather than introducing new optical modalities, yet have had a measureable impact on the field by enabling broader community participation. Altair-LSFM follows this tradition, providing sub-cellular resolution performance comparable to advanced systems like LLSM, while emphasizing reproducibility, ease of construction through a precision-machined baseplate, and comprehensive documentation to facilitate dissemination and adoption.

(2) Using similar objectives (Nikon 25x and Thorlabs 20x), the results obtained are similar to those of the LLSM system (using a Gaussian beam without laser modulation). However, the article does not mention the difficulties of mounting the sample in the implemented configuration.

We appreciate the reviewer’s comment and agree that there are practical challenges associated with handling 5 mm diameter coverslips in this configuration. In the revised manuscript, we now explicitly describe these challenges and provide practical solutions. Specifically, we highlight the use of a custommachined coverslip holder designed to simplify mounting and handling, and we direct readers to an alternative configuration using the Zeiss W Plan-Apochromat 20×/1.0 objective, which eliminates the need for small coverslips altogether.

(3) The authors present a low-cost, open-source system. Although they provide open source code for the software (navigate), the use of proprietary electronics (ASI, NI, etc.) makes the system relatively expensive. Its low cost is not justified.

We appreciate the reviewer’s perspective and understand the concern regarding the use of proprietary control hardware such as the ASI Tiger Controller and NI data acquisition cards. Our decision to use these components was intentional: relying on a unified, professionally supported and maintained platform minimizes complexity associated with sourcing, configuring, and integrating hardware from multiple vendors, thereby reducing non-financial barriers to entry for non-specialist users.

Importantly, these components are not the primary cost driver of Altair-LSFM (they represent roughly 18% of the total system cost). Nonetheless, for individuals where the price is prohibitive, we also outline several viable cost-reduction options in the revised manuscript (e.g., substituting manual stages, omitting the filter wheel, or using industrial CMOS cameras), while discussing the trade-offs these substitutions introduce in performance and usability. These considerations are now summarized in Supplementary Note 1, which provides a transparent rationale for our design and cost decisions.

Finally, we note that even with these professional-grade components, Altair-LSFM remains substantially less expensive than commercial systems offering comparable optical performance, such as LLSM implementations from Zeiss or 3i.

(4) The fibroblast images provided are of exceptional quality. However, these are fixed samples. The system lacks the necessary elements for monitoring cells in vivo, such as temperature or pH control.

We thank the reviewer for their positive comment regarding the quality of our data. As noted, the current manuscript focuses on validating the optical performance and resolution of the system using fixed specimens to ensure reproducibility and stability.

We fully agree on the importance of environmental control for live-cell imaging. In the revised manuscript, we now describe in detail how temperature regulation can be achieved using a custom-designed heated sample chamber, accompanied by detailed assembly instructions on our GitHub repository and summarized in Supplementary Note 2. For pH stabilization in systems lacking a 5% CO₂ atmosphere, we recommend supplementing the imaging medium with 10–25 mM HEPES buffer. Additionally, we include new live-cell imaging data demonstrating that Altair-LSFM supports in vitro time-lapse imaging of dynamic cellular processes under controlled temperature conditions.

**Reviewer #2 (Public review):**
Summary:The authors present Altair-LSFM (Light Sheet Fluorescence Microscope), a high-resolution, open-source microscope, that is relatively easy to align and construct and achieves sub-cellular resolution. The authors developed this microscope to fill a perceived need that current open-source systems are primarily designed for large specimens and lack sub-cellular resolution or are difficult to construct and align, and are not stable. While commercial alternatives exist that offer sub-cellular resolution, they are expensive. The authors' manuscript centers around comparisons to the highly successful lattice light-sheet microscope, including the choice of detection and excitation objectives. The authors thus claim that there remains a critical need for high-resolution, economical, and easy-to-implement LSFM systems.

We thank the reviewer for their thoughtful summary. We agree that existing open-source systems primarily emphasize imaging of large specimens, whereas commercial systems that achieve sub-cellular resolution remain costly and complex. Our aim with Altair-LSFM was to bridge this gap—providing LLSM-level performance in a substantially more accessible and reproducible format. By combining high-NA optics with a precision-machined baseplate and open-source documentation, Altair offers a practical, high-resolution solution that can be readily adopted by non-specialist laboratories.

Strengths:The authors succeed in their goals of implementing a relatively low-cost (~ USD 150K) open-source microscope that is easy to align. The ease of alignment rests on using custom-designed baseplates with dowel pins for precise positioning of optics based on computer analysis of opto-mechanical tolerances, as well as the optical path design. They simplify the excitation optics over Lattice light-sheet microscopes by using a Gaussian beam for illumination while maintaining lateral and axial resolutions of 235 and 350 nm across a 260-um field of view after deconvolution. In doing so they rest on foundational principles of optical microscopy that what matters for lateral resolution is the numerical aperture of the detection objective and proper sampling of the image field on to the detection, and the axial resolution depends on the thickness of the light-sheet when it is thinner than the depth of field of the detection objective. This concept has unfortunately not been completely clear to users of high-resolution light-sheet microscopes and is thus a valuable demonstration. The microscope is controlled by an open-source software, Navigate, developed by the authors, and it is thus foreseeable that different versions of this system could be implemented depending on experimental needs while maintaining easy alignment and low cost. They demonstrate system performance successfully by characterizing their sheet, point-spread function, and visualization of sub-cellular structures in mammalian cells, including microtubules, actin filaments, nuclei, and the Golgi apparatus.

We thank the reviewer for their thoughtful and generous assessment of our work. We are pleased that the manuscript’s emphasis on fundamental optical principles, design rationale, and practical implementation was clearly conveyed. We agree that Altair’s modular and accessible architecture provides a strong foundation for future variants tailored to specific experimental needs. To facilitate this, we have made all Zemax simulations, CAD files, and build documentation openly available on our GitHub repository, enabling users to adapt and extend the system for diverse imaging applications.

Weaknesses:There is a fixation on comparison to the first-generation lattice light-sheet microscope, which has evolved significantly since then:(1) The authors claim that commercial lattice light-sheet microscopes (LLSM) are "complex, expensive, and alignment intensive", I believe this sentence applies to the open-source version of LLSM, which was made available for wide dissemination. Since then, a commercial solution has been provided by 3i, which is now being used in multiple cores and labs but does require routine alignments. However, Zeiss has also released a commercial turn-key system, which, while expensive, is stable, and the complexity does not interfere with the experience of the user. Though in general, statements on ease of use and stability might be considered anecdotal and may not belong in a scientific article, unreferenced or without data.

We thank the reviewer for this thoughtful and constructive comment. We have revised the manuscript to more clearly distinguish between the original open-source implementation of LLSM and subsequent commercial versions by 3i and ZEISS. The revised Introduction and Discussion now explicitly note that while open-source and early implementations of LLSM can require expert alignment and maintenance, commercial systems—particularly the ZEISS Lattice Lightsheet 7—are designed for automated operation and stable, turn-key use, albeit at higher cost and with limited modifiability. We have also moderated earlier language regarding usability and stability to avoid anecdotal phrasing.

We also now provide a more objective proxy for system complexity: the number of optical elements that require precise alignment during assembly and maintenance thereafter. The original open-source LLSM setup includes approximately 29 optical components that must each be carefully positioned laterally, angularly, and coaxially along the optical path. In contrast, the first-generation Altair-LSFM system contains only nine such elements. By this metric, Altair-LSFM is considerably simpler to assemble and align, supporting our overarching goal of making high-resolution light-sheet imaging more accessible to non-specialist laboratories.

(2) One of the major limitations of the first generation LLSM was the use of a 5 mm coverslip, which was a hinderance for many users. However, the Zeiss system elegantly solves this problem, and so does Oblique Plane Microscopy (OPM), while the Altair-LSFM retains this feature, which may dissuade widespread adoption. This limitation and how it may be overcome in future iterations is not discussed.

We thank the reviewer for this helpful comment. We agree that the use of 5 mm diameter coverslips, while enabling high-NA imaging in the current Altair-LSFM configuration, may pose a practical limitation for some users. We now discuss this more explicitly in the revised manuscript. Specifically, we note that replacing the detection objective provides a straightforward solution to this constraint. For example, as demonstrated by Moore et al. (Lab Chip, 2021), pairing the Zeiss W Plan-Apochromat 20×/1.0 detection objective with the Thorlabs TL20X-MPL illumination objective allows imaging beyond the physical surfaces of both objectives, eliminating the need for small-format coverslips. In the revised text, we propose this modification as an accessible path toward greater compatibility with conventional sample mounting formats. We also note in the Discussion that Oblique Plane Microscopy (OPM) inherently avoids such nonstandard mounting requirements and, owing to its single-objective architecture, is fully compatible with standard environmental chambers.

(3) Further, on the point of sample flexibility, all generations of the LLSM, and by the nature of its design, the OPM, can accommodate live-cell imaging with temperature, gas, and humidity control. It is unclear how this would be implemented with the current sample chamber. This limitation would severely limit use cases for cell biologists, for which this microscope is designed. There is no discussion on this limitation or how it may be overcome in future iterations.

We thank the reviewer for this important observation and agree that environmental control is critical for live-cell imaging applications. It is worth noting that the original open-source LLSM design, as well as the commercial version developed by 3i, provided temperature regulation but did not include integrated control of CO2 or humidity. Despite this limitation, these systems have been widely adopted and have generated significant biological insights. We also acknowledge that both OPM and the ZEISS implementation of LLSM offer clear advantages in this respect, providing compatibility with standard commercial environmental chambers that support full regulation of temperature, CO₂, and humidity.

In the revised manuscript, we expand our discussion of environmental control in Supplementary Note 2, where we describe the Altair-LSFM chamber design in more detail and discuss its current implementation of temperature regulation and HEPES-based pH stabilization. Additionally, the Discussion now explicitly notes that OPM avoids the challenges associated with non-standard sample mounting and is inherently compatible with conventional environmental enclosures.

(4) The authors' comparison to LLSM is constrained to the "square" lattice, which, as they point out, is the most used optical lattice (though this also might be considered anecdotal). The LLSM original design, however, goes far beyond the square lattice, including hexagonal lattices, the ability to do structured illumination, and greater flexibility in general in terms of light-sheet tuning for different experimental needs, as well as not being limited to just sample scanning. Thus, the Alstair-LSFM cannot compare to the original LLSM in terms of versatility, even if comparisons to the resolution provided by the square lattice are fair.

We agree that the original LLSM design offers substantially greater flexibility than what is reflected in our initial comparison, including the ability to generate multiple lattice geometries (e.g., square and hexagonal), operate in structured illumination mode, and acquire volumes using both sample- and lightsheet–scanning strategies. To address this, we now include Supplementary Note 3 that provides a detailed overview of the illumination modes and imaging flexibility afforded by the original LLSM implementation, and how these capabilities compare to both the commercial ZEISS Lattice Lightsheet 7 and our AltairLSFM system. In addition, we have revised the discussion to explicitly acknowledge that the original LLSM could operate in alternative scan strategies beyond sample scanning, providing greater context for readers and ensuring a more balanced comparison.

(5) There is no demonstration of the system's live-imaging capabilities or temporal resolution, which is the main advantage of existing light-sheet systems.

In the revised manuscript, we now include a demonstration of live-cell imaging to directly validate AltairLSFM’s suitability for dynamic biological applications. We also explicitly discuss the temporal resolution of the system in the main text (see Optoelectronic Design of Altair-LSFM), where we detail both software- and hardware-related limitations. Specifically, we evaluate the maximum imaging speed achievable with Altair-LSFM in conjunction with our open-source control software, navigate.

For simplicity and reduced optoelectronic complexity, the current implementation powers the piezo through the ASI Tiger Controller, which modestly reduces its bandwidth. Nonetheless, for a 100 µm stroke typical of light-sheet imaging, we achieved sufficient performance to support volumetric imaging at most biologically relevant timescales. These results, along with additional discussion of the design trade-offs and performance considerations, are now included in the revised manuscript and expanded upon in the supplementary material.

While the microscope is well designed and completely open source, it will require experience with optics, electronics, and microscopy to implement and align properly. Experience with custom machining or soliciting a machine shop is also necessary. Thus, in my opinion, it is unlikely to be implemented by a lab that has zero prior experience with custom optics or can hire someone who does. Altair-LSFM may not be as easily adaptable or implementable as the authors describe or perceive in any lab that is interested, even if they can afford it. The authors indicate they will offer "workshops," but this does not necessarily remove the barrier to entry or lower it, perhaps as significantly as the authors describe.

We appreciate the reviewer’s perspective and agree that building any high-performance custom microscope—Altair-LSFM included—requires a basic understanding of (or willingness to learn) optics, electronics, and instrumentation. Such a barrier exists for all open-source microscopes, and our goal is not to eliminate this requirement entirely but to substantially reduce the technical and logistical challenges that typically accompany the construction of custom light-sheet systems.

Importantly, no machining experience or in-house fabrication capabilities are required. Users can simply submit the provided CAD design files and specifications directly to commercial vendors for fabrication. We have made this process as straightforward as possible by supplying detailed build instructions, recommended materials, and vendor-ready files through our GitHub repository. Our dissemination strategy draws inspiration from other successful open-source projects such as mesoSPIM, which has seen widespread adoption—over 30 implementations worldwide—through a similar model of exhaustive documentation, open-source software, and community support via user meetings and workshops.

We also recognize that documentation alone cannot fully replace hands-on experience. To further lower barriers to adoption, we are actively working with commercial vendors to streamline procurement and assembly, and Altair-LSFM is supported by a Biomedical Technology Development and Dissemination (BTDD) grant that provides resources for hosting workshops, offering real-time community support, and developing supplementary training materials.

In the revised manuscript, we now expand the Discussion to explicitly acknowledge these implementation considerations and to outline our ongoing efforts to support a broad and diverse user base, ensuring that laboratories with varying levels of technical expertise can successfully adopt and maintain the Altair-LSFM platform.

There is a claim that this design is easily adaptable. However, the requirement of custom-machined baseplates and in silico optimization of the optical path basically means that each new instrument is a new design, even if the Navigate software can be used. It is unclear how Altair-LSFM demonstrates a modular design that reduces times from conception to optimization compared to previous implementations.

We thank the reviewer for this insightful comment and agree that our original language regarding adaptability may have overstated the degree to which Altair-LSFM can be modified without prior experience. It was not our intention to imply that the system can be easily redesigned by users with limited technical background. Meaningful adaptations of the optical or mechanical design do require expertise in optical layout, optomechanical design, and alignment.

That said, for laboratories with such expertise, we aim to facilitate modifications by providing comprehensive resources—including detailed Zemax simulations, complete CAD models, and alignment documentation. These materials are intended to reduce the development burden for expert users seeking to tailor the system to specific experimental requirements, without necessitating a complete re-optimization of the optical path from first principles.

In the revised manuscript, we clarify this point and temper our language regarding adaptability to better reflect the realistic scope of customization. Specifically, we now state in the Discussion: “For expert users who wish to tailor the instrument, we also provide all Zemax illumination-path simulations and CAD files, along with step-by-step optimization protocols, enabling modification and re-optimization of the optical system as needed.” This revision ensures that readers clearly understand that Altair-LSFM is designed for reproducibility and straightforward assembly in its default configuration, while still offering the flexibility for modification by experienced users.

**Reviewer #3 (Public review):**
Summary:This manuscript introduces a high-resolution, open-source light-sheet fluorescence microscope optimized for sub-cellular imaging. The system is designed for ease of assembly and use, incorporating a custommachined baseplate and in silico optimized optical paths to ensure robust alignment and performance. The authors demonstrate lateral and axial resolutions of ~235 nm and ~350 nm after deconvolution, enabling imaging of sub-diffraction structures in mammalian cells. The important feature of the microscope is the clever and elegant adaptation of simple gaussian beams, smart beam shaping, galvo pivoting and high NA objectives to ensure a uniform thin light-sheet of around 400 nm in thickness, over a 266 micron wide Field of view, pushing the axial resolution of the system beyond the regular diffraction limited-based tradeoffs of light-sheet fluorescence microscopy. Compelling validation using fluorescent beads and multicolor cellular imaging highlights the system's performance and accessibility. Moreover, a very extensive and comprehensive manual of operation is provided in the form of supplementary materials. This provides a DIY blueprint for researchers who want to implement such a system.

We thank the reviewer for their thoughtful and positive assessment of our work. We appreciate their recognition of Altair-LSFM’s design and performance, including its ability to achieve high-resolution, imaging throughout a 266-micron field of view. While Altair-LSFM approaches the practical limits of diffraction-limited performance, it does not exceed the fundamental diffraction limit; rather, it achieves near-theoretical resolution through careful optical optimization, beam shaping, and alignment. We are grateful for the reviewer’s acknowledgment of the accessibility and comprehensive documentation that make this system broadly implementable.

Strengths:(1) Strong and accessible technical innovation: With an elegant combination of beam shaping and optical modelling, the authors provide a high-resolution light-sheet system that overcomes the classical light-sheet tradeoff limit of a thin light-sheet and a small field of view. In addition, the integration of in silico modelling with a custom-machined baseplate is very practical and allows for ease of alignment procedures. Combining these features with the solid and super-extensive guide provided in the supplementary information, this provides a protocol for replicating the microscope in any other lab.(2) Impeccable optical performance and ease of mounting of samples: The system takes advantage of the same sample-holding method seen already in other implementations, but reduces the optical complexity.At the same time, the authors claim to achieve similar lateral and axial resolution to Lattice-light-sheet microscopy although without a direct comparison (see below in the "weaknesses" section). The optical characterization of the system is comprehensive and well-detailed. Additionally, the authors validate the system imaging sub-cellular structures in mammalian cells.(3) Transparency and comprehensiveness of documentation and resources: A very detailed protocol provides detailed documentation about the setup, the optical modeling, and the total cost.

We thank the reviewer for their thoughtful and encouraging comments. We are pleased that the technical innovation, optical performance, and accessibility of Altair-LSFM were recognized. Our goal from the outset was to develop a diffraction-limited, high-resolution light-sheet system that balances optical performance with reproducibility and ease of implementation. We are also pleased that the use of precisionmachined baseplates was recognized as a practical and effective strategy for achieving performance while maintaining ease of assembly.

Weaknesses:(1) Limited quantitative comparisons: Although some qualitative comparison with previously published systems (diSPIM, lattice light-sheet) is provided throughout the manuscript, some side-by-side comparison would be of great benefit for the manuscript, even in the form of a theoretical simulation. While having a direct imaging comparison would be ideal, it's understandable that this goes beyond the interest of the paper; however, a table referencing image quality parameters (taken from the literature), such as signalto-noise ratio, light-sheet thickness, and resolutions, would really enhance the features of the setup presented. Moreover, based also on the necessity for optical simplification, an additional comment on the importance/difference of dual objective/single objective light-sheet systems could really benefit the discussion.

In the revised manuscript, we have significantly expanded our discussion of different light-sheet systems to provide clearer quantitative and conceptual context for Altair-LSFM. These comparisons are based on values reported in the literature, as we do not have access to many of these instruments (e.g., DaXi, diSPIM, or commercial and open-source variants of LLSM), and a direct experimental comparison is beyond the scope of this work.

We note that while quantitative parameters such as signal-to-noise ratio are important, they are highly sample-dependent and strongly influenced by imaging conditions, including fluorophore brightness, camera characteristics, and filter bandpass selection. For this reason, we limited our comparison to more general image-quality metrics—such as light-sheet thickness, resolution, and field of view—that can be reliably compared across systems.

Finally, per the reviewer’s recommendation, we have added additional discussion clarifying the differences between dual-objective and single-objective light-sheet architectures, outlining their respective strengths, limitations, and suitability for different experimental contexts.

(2) Limitation to a fixed sample: In the manuscript, there is no mention of incubation temperature, CO₂ regulation, Humidity control, or possible integration of commercial environmental control systems. This is a major limitation for an imaging technique that owes its popularity to fast, volumetric, live-cell imaging of biological samples.

We fully agree that environmental control is critical for live-cell imaging applications. In the revised manuscript, we now describe the design and implementation of a temperature-regulated sample chamber in Supplementary Note 2, which maintains stable imaging conditions through the use of integrated heating elements and thermocouples. This approach enables precise temperature control while minimizing thermal gradients and optical drift. For pH stabilization, we recommend the use of 10–25 mM HEPES in place of CO₂ regulation, consistent with established practice for most light-sheet systems, including the initial variant of LLSM. Although full humidity and CO₂ control are not readily implemented in dual-objective configurations, we note that single-objective designs such as OPM are inherently compatible with commercial environmental chambers and avoid these constraints. Together, these additions clarify how environmental control can be achieved within Altair-LSFM and situate its capabilities within the broader LSFM design space.

(3) System cost and data storage cost: While the system presented has the advantage of being opensource, it remains relatively expensive (considering the 150k without laser source and optical table, for example). The manuscript could benefit from a more direct comparison of the performance/cost ratio of existing systems, considering academic settings with budgets that most of the time would not allow for expensive architectures. Moreover, it would also be beneficial to discuss the adaptability of the system, in case a 30k objective could not be feasible. Will this system work with different optics (with the obvious limitations coming with the lower NA objective)? This could be an interesting point of discussion. Adaptability of the system in case of lower budgets or more cost-effective choices, depending on the needs.

We agree that cost considerations are critical for adoption in academic environments. We would also like to clarify that the quoted $150k includes the optical table and laser source. In the revised manuscript, Supplementary Note 1 now includes an expanded discussion of cost–performance trade-offs and potential paths for cost reduction.

Last, not much is said about the need for data storage. Light-sheet microscopy's bottleneck is the creation of increasingly large datasets, and it could be beneficial to discuss more about the storage needs and the quantity of data generated.

In the revised manuscript, we now include Supplementary Note 4, which provides a high-level discussion of data storage needs, approximate costs, and practical strategies for managing large datasets generated by light-sheet microscopy. This section offers general guidance—including file-format recommendations, and cost considerations—but we note that actual costs will vary by institution and contractual agreements.

Conclusion:Altair-LSFM represents a well-engineered and accessible light-sheet system that addresses a longstanding need for high-resolution, reproducible, and affordable sub-cellular light-sheet imaging. While some aspects-comparative benchmarking and validation, limitation for fixed samples-would benefit from further development, the manuscript makes a compelling case for Altair-LSFM as a valuable contribution to the open microscopy scientific community.
**Recommendations for the authors:**

**Reviewer #2 (Recommendations for the authors):**
(1) A picture, or full CAD design of the complete instrument, should be included as a main figure.

A complete CAD rendering of the microscope is now provided in Supplementary Figure 4.

(2) There is no quantitative comparison of the effects of the tilting resonant galvo; only a cartoon, a figure should be included.

The cartoon was intended purely as an educational illustration to conceptually explain the role of the tilting resonant galvo in shaping and homogenizing the light sheet. To clarify this intent, we have revised both the figure legend and corresponding text in the main manuscript. For readers seeking quantitative comparisons, we now reference the original study that provides a detailed analysis of this optical approach, as well as a review on the subject.

(3) Description of L4 is missing in the Figure 1 caption.

Thank you for catching this omission. We have corrected it.

(4) The beam profiles in Figures 1c and 3a, please crop and make the image bigger so the profile can be appreciated. The PSFs in Figure 3c-e should similarly be enlarged and presented using a dynamic range/LUT such that any aberrations can be appreciated.

In Figure 1c, our goal was to qualitatively illustrate the uniformity of the light-sheet across the full field of view, while Figure 1d provided the corresponding quantitative cross-section. To improve clarity, we have added an additional figure panel offering a higher-magnification, localized view of the light-sheet profile. For Figure 3c–e, we have enlarged the PSF images and adjusted the display range to better convey the underlying signal and allow subtle aberrations to be appreciated.

(5) It is unclear why LLSM is being used as the gold standard, since in its current commercial form, available from Zeiss, it is a turn-key system designed for core facilities. The original LLSM is also a versatile instrument that provides much more than the square lattice for illumination, including structured illumination, hexagonal lattices, live-cell imaging, wide-field illumination, different scan modes, etc. These additional features are not even mentioned when compared to the Altair-LSFM. If a comparison is to be provided, it should be fair and balanced. Furthermore, as outlined in the public review, anecdotal statements on "most used", "difficult to align", or "unstable" should not be provided without data.

In the revised manuscript, we have carefully removed anecdotal statements and, where appropriate, replaced them with quantitative or verifiable information. For instance, we now explicitly report that the square lattice was used in 16 of the 20 figure subpanels in the original LLSM publication, and we include a proxy for optical complexity based on the number of optical elements requiring alignment in each system.

We also now clearly distinguish between the original LLSM design—which supports multiple illumination and scanning modes—and its subsequent commercial variants, including the ZEISS Lattice Lightsheet 7, which prioritizes stability and ease of use over configurational flexibility (see Supplementary Note 3).

(6) The authors should recognize that implementing custom optics, no matter how well designed, is a big barrier to cross for most cell biology labs.

We fully understand and now acknowledge in the main text that implementing custom optics can present a significant barrier, particularly for laboratories without prior experience in optical system assembly. However, similar challenges were encountered during the adoption of other open-source microscopy platforms, such as mesoSPIM and OpenSPIM, both of which have nonetheless achieved widespread implementation. Their success has largely been driven by exhaustive documentation, strong community support, and standardized design principles—approaches we have also prioritized in Altair-LSFM. We have therefore made all CAD files, alignment guides, and detailed build documentation publicly available and continue to develop instructional materials and community resources to further reduce the barrier to adoption.

(7) Statements on "hands on workshops" though laudable, may not be appropriate to include in a scientific publication without some documentation on the influence they have had on implanting the microscope.

We understand the concern. Our intention in mentioning hands-on workshops was to convey that the dissemination effort is supported by an NIH Biomedical Technology Development and Dissemination grant, which includes dedicated channels for outreach and community engagement. Nonetheless, we agree that such statements are not appropriate without formal documentation of their impact, and we have therefore removed this text from the revised manuscript.

(8) It is claimed that the microscope is "reliable" in the discussion, but with no proof, long-term stability should be assessed and included.

Our experience with Altair-LSFM has been that it remains well-aligned over time—especially in comparison to other light-sheet systems we worked on throughout the last 11 years—we acknowledge that this assessment is anecdotal. As such, we have omitted this claim from the revised manuscript.

(9) Due to the reliance on anecdotal statements and comparisons without proof to other systems, this paper at times reads like a brochure rather than a scientific publication. The authors should consider editing their manuscript accordingly to focus on the technical and quantifiable aspects of their work.

We agree with the reviewer’s assessment and have revised the manuscript to remove anecdotal comparisons and subjective language. Where possible, we now provide quantitative metrics or verifiable data to support our statements.

**Reviewer #3 (Recommendations for the authors):**
Other minor points that could improve the manuscript (although some of these points are explained in the huge supplementary manual):(1) The authors explain thoroughly their design, and they chose a sample-scanning method. I think that a brief discussion of the advantages and disadvantages of such a method over, for example, a laserscanning system (with fixed sample) in the main text will be highly beneficial for the users.

In the revised manuscript, we now include a brief discussion in the main text outlining the advantages and limitations of a sample-scanning approach relative to a light-sheet–scanning system. Specifically, we note that for thin, adherent specimens, sample scanning minimizes the optical path length through the sample, allowing the use of more tightly focused illumination beams that improve axial resolution. We also include a new supplementary figure illustrating how this configuration reduces the propagation length of the illumination light sheet, thereby enhancing axial resolution.

(2) The authors justify selecting a 0.6 NA illumination objective over alternatives (e.g., Special Optics), but the manuscript would benefit from a more quantitative trade-off analysis (beam waist, working distance, sample compatibility) with other possibilities. Within the objective context, a comparison of the performances of this system with the new and upcoming single-objective light-sheet methods (and the ones based also on optical refocusing, e.g., DAXI) would be very interesting for the goodness of the manuscript.

In the revised manuscript, we now provide a quantitative trade-off analysis of the illumination objectives in Supplementary Note 1, including comparisons of beam waist, working distance, and sample compatibility. This section also presents calculated point spread functions for both the 0.6 NA and 0.67 NA objectives, outlining the performance trade-offs that informed our design choice. In addition, Supplementary Note 3 now includes a broader comparison of Altair-LSFM with other light-sheet modalities, including diSPIM, ASLM, and OPM, to further contextualize the system’s capabilities within the evolving light-sheet microscopy landscape.

(3) The modularity of the system is implied in the context of the manuscript, but not fully explained. The authors should specify more clearly, for example, if cameras could be easily changed, objectives could be easily swapped, light-sheet thickness could be tuned by changing cylindrical lens, how users might adapt the system for different samples (e.g., embryos, cleared tissue, live imaging), .etc, and discuss eventual constraints or compatibility issues to these implementations.

Altair-LSFM was explicitly designed and optimized for imaging live adherent cells, where sample scanning and short light-sheet propagation lengths provide optimal axial resolution (Supplementary Note 3). While the same platform could be used for superficial imaging in embryos, systems implementing multiview illumination and detection schemes are better suited for such specimens. Similarly, cleared tissue imaging typically requires specialized solvent-compatible objectives and approaches such as ASLM that maximize the field of view. We have now added some text to the Design Principles section that explicitly state this.

Altair-LSFM offers varying levels of modularity depending on the user’s level of expertise. For entry-level users, the illumination numerical aperture—and therefore the light-sheet thickness and propagation length—can be readily adjusted by tuning the rectangular aperture conjugate to the back pupil of the illumination objective, as described in the Design Principles section. For mid-level users, alternative configurations of Altair-LSFM, including different detection objectives, stages, filter wheels, or cameras, can be readily implemented (Supplementary Note 1). Importantly, navigate natively supports a broad range of hardware devices, and new components can be easily integrated through its modular interface. For expert users, all Zemax simulations, CAD models, and step-by-step optimization protocols are openly provided, enabling complete re-optimization of the optical design to meet specific experimental requirements.

(4) Resolution measurements before and after deconvolution are central to the performance claim, but the deconvolution method (PetaKit5D) is only briefly mentioned in the main text, it's not referenced, and has to be clarified in more detail, coherently with the precision of the supplementary information. More specifically, PetaKit5D should be referenced in the main text, the details of the deconvolution parameters discussed in the Methods section, and the computational requirements should also be mentioned.

In the revised manuscript, we now provide a dedicated description of the deconvolution process in the Methods section, including the specific parameters and algorithms used. We have also explicitly referenced PetaKit5D in the main text to ensure proper attribution and clarity. Additionally, we note the computational requirements associated with this analysis in the same section for completeness.

(5) Image post-processing is not fully explained in the main text. Since the system is sample-scanning based, no word in the main text is spent on deskewing, which is an integral part of the post-processing to obtain a "straight" 3D stack. Since other systems implement such a post-processing algorithm (for example, single-objective architectures), it would be beneficial to have some discussion about this, and also a brief comparison to other systems in the main text in the methods section.

In the revised manuscript, we now explicitly describe both deskewing (shearing) and deconvolution procedures in the Alignment and Characterization section of the main text and direct readers to the Methods section. We also briefly explain why the data must be sheared to correct for the angled sample-scanning geometry for LLSM and Altair-LSFM, as well as both sample-scanning and laser-scanning-variants of OPMs.

(6) A brief discussion on comparative costs with other systems (LLSM, dispim, etc.) could be helpful for non-imaging expert researchers who could try to implement such an optical architecture in their lab.

Unfortunately, the exact costs of commercial systems such as LLSM or diSPIM are typically not publicly available, as they depend on institutional agreements and vendor-specific quotations. Nonetheless, we now provide approximate cost estimates in Supplementary Note 1 to help readers and prospective users gauge the expected scale of investment relative to other advanced light-sheet microscopy systems.

(7) The "navigate" control software is provided, but a brief discussion on its advantages compared to an already open-access system, such as Micromanager, could be useful for the users.

In the revised manuscript, we now include Supplementary Note 5 that discusses the advantages and disadvantages of different open-source microscope control platforms, including navigate and MicroManager. In brief, navigate was designed to provide turnkey support for multiple light-sheet architectures, with pre-configured acquisition routines optimized for Altair-LSFM, integrated data management with support for multiple file formats (TIFF, HDF5, N5, and Zarr), and full interoperability with OMEcompliant workflows. By contrast, while Micro-Manager offers a broader library of hardware drivers, it typically requires manual configuration and custom scripting for advanced light-sheet imaging workflows.

(8) The cost and parts are well documented, but the time and expertise required are not crystal clear.Adding a simple time estimate (perhaps in the Supplement Section) of assembly/alignment/installation/validation and first imaging will be very beneficial for users. Also, what level of expertise is assumed (prior optics experience, for example) to be needed to install a system like this? This can help non-optics-expert users to better understand what kind of adventure they are putting themselves through.

We thank the reviewer for this helpful suggestion. To address this, we have added Supplementary Table S5, which provides approximate time estimates for assembly, alignment, validation, and first imaging based on the user’s prior experience with optical systems. The table distinguishes between novice (no prior experience), moderate (some experience using but not assembling optical systems), and expert (experienced in building and aligning optical systems) users. This addition is intended to give prospective builders a realistic sense of the time commitment and level of expertise required to assemble and validate AltairLSFM.

Minor things in the main text:(1) Line 109: The cost is considered "excluding the laser source". But then in the table of costs, you mention L4cc as a "multicolor laser source", for 25 K. Can you explain this better? Are the costs correct with or without the laser source?

We acknowledge that the statement in line 109 was incorrect—the quoted ~$150k system cost does include the laser source (L4cc, listed at $25k in the cost table). We have corrected this in the revised manuscript.

(2) Line 113: You say "lateral resolution, but then you state a 3D resolution (230 nm x 230 nm x 370 nm). This needs to be fixed.

Thank you, we have corrected this.

(3) Line 138: Is the light-sheet uniformity proven also with a fluorescent dye? This could be beneficial for the main text, showing the performance of the instrument in a fluorescent environment.

The light-sheet profiles shown in the manuscript were acquired using fluorescein to visualize the beam. We have revised the main text and figure legends to clearly state this.

(4) Line 149: This is one of the most important features of the system, defying the usual tradeoff between light-sheet thickness and field of view, with a regular Gaussian beam. I would clarify more specifically how you achieve this because this really is the most powerful takeaway of the paper.

We thank the reviewer for this key observation. The ability of Altair-LSFM to maintain a thin light sheet across a large field of view arises from diffraction effects inherent to high NA illumination. Specifically, diffraction elongates the PSF along the beam’s propagation direction, effectively extending the region over which the light sheet remains sufficiently thin for high-resolution imaging. This phenomenon, which has been the subject of active discussion within the light-sheet microscopy community, allows Altair-LSFM to partially overcome the conventional trade-off between light-sheet thickness and propagation length. We now clarify this point in the main text and provide a more detailed discussion in Supplementary Note 3, which is explicitly referenced in the discussion of the revised manuscript.

(5) Line 171: You talk about repeatable assembly...have you tried many different baseplates? Otherwise, this is a complicated statement, since this is a proof-of-concept paper.

We thank the reviewer for this comment. We have not yet validated the design across multiple independently assembled baseplates and therefore agree that our previous statement regarding repeatable assembly was premature. To avoid overstating the current level of validation, we have removed this statement from the revised manuscript.

(6) Line 187: same as above. You mention "long-term stability". For how long did you try this? This should be specified in numbers (days, weeks, months, years?) Otherwise, it is a complicated statement to make, since this is a proof-of-concept paper.

We also agree that referencing long-term stability without quantitative backing is inappropriate, and have removed this statement from the revised manuscript.

(7) Line 198: "rapid z-stack acquisition. How rapid? Also, what is the limitation of the galvo-scanning in terms of the imaging speed of the system? This should be noted in the methods section.

In the revised manuscript, we now clarify these points in the Optoelectronic Design section. Specifically, we explicitly note that the resonant galvo used for shadow reduction operates at 4 kHz, ensuring that it is not rate-limiting for any imaging mode. In the same section, we also evaluate the maximum acquisition speeds achievable using navigate and report the theoretical bandwidth of the sample-scanning piezo, which together define the practical limits of volumetric acquisition speed for Altair-LSFM.

(8) Line 234: Peta5Kit is discussed in the additional documentation, but should be referenced here, as well.

We now reference and cite PetaKit5D.

(9) Line 256: "values are on par with LLSM", but no values are provided. Some details should also be provided in the main text.

In the revised manuscript, we now provide the lateral and axial resolution values originally reported for LLSM in the main text to facilitate direct comparison with Altair-LSFM. Additionally, Supplementary Note 3 now includes an expanded discussion on the nuances of resolution measurement and reporting in lightsheet microscopy.

Figures:(1) Figure 1 could be implemented with Figure 3. They're both discussing the validation of the system (theoretically and with simulations), and they could be together in different panels of the same figure. The experimental light-sheet seems to be shown in a transmission mode. Showing a pattern in a fluorescent dye could also be beneficial for the paper.

In Figure 1, our goal was to guide readers through the design process—illustrating how the detection objective’s NA sets the system’s resolution, which defines the required pixel size for Nyquist sampling and, in turn, the field of view. We then use Figure 1b–c to show how the illumination beam was designed and simulated to achieve that field of view. In contrast, Figure 3 presents the experimental validation of the illumination system. To avoid confusion, we now clarify in the text that the light sheet shown in Figure 3 was visualized in a fluorescein solution and imaged in transmission mode. While we agree that Figures 1 and 3 both serve to validate the system, we prefer to keep them as separate figures to maintain focus within each panel. We believe this organization better supports the narrative structure and allows readers to digest the theoretical and experimental validations independently.

(2) Figure 3: Panels d and e show the same thing. Why would you expect that xz and yz profiles should be different? Is this due to the orientation of the objectives towards the sample?

In Figure 3, we present the PSF from all three orthogonal views, as this provides the most transparent assessment of PSF quality—certain aberration modes can be obscured when only select perspectives are shown. In principle, the XZ and YZ projections should be equivalent in a well-aligned system. However, as seen in the XZ projection, a small degree of coma is present that is not evident in the YZ view. We now explicitly note this observation in the revised figure caption to clarify the difference between these panels.

(3) Figure 4's single boxes lack a scale bar, and some of the Supplementary Figures (e.g. Figure 5) lack detailed axis labels or scale bars. Also, in the detailed documentation, some figures are referred to as Figure 5. Figure 7 or, for example, figure 6. Figure 8, and this makes the cross-references very complicated to follow

In the revised manuscript, we have corrected these issues. All figures and supplementary figures now include appropriate scale bars, axis labels, and consistent formatting. We have also carefully reviewed and standardized all cross-references throughout the main text and supplementary documentation to ensure that figure numbering is accurate and easy to follow.